# Limitations of the 1% experiment as the benchmark idealized experiment for carbon cycle intercomparison in $C^4MIP$

Andrew H. MacDougall[1]

[1]Climate & Environment, St. Francis Xavier University, Antigonish, Nova Scotia, Canada.

**Correspondence:** AH MacDougall (amacdoug@stfx.ca)

**Abstract.** Idealized climate change simulations are used as benchmark experiments to facilitate the comparison of ensembles of climate models. In the Fifth phase of the Climate Model Intercomparison Project (CMIP5) the 1% per yearly compounded change in atmospheric $CO_2$ concentration experiment was used to compare Earth System Models with full representations of the global carbon cycle ($C^4MIP$). However this "1% experiment" was never intended for such a purpose and implies a rise in atmospheric $CO_2$ concentration at double the rate of the instrumental record. Here we examine this choice by using an intermediate complexity climate model to compare the 1% experiment to an idealized $CO_2$ pathway derived from a logistic function. The comparison shows three key differences in model output when forcing the model with the Logistic experiment. (1) The model forced with the logistic experiment exhibits a transition of the land biosphere from a carbon sink to a carbon source, a feature absent when forcing the model with the 1% experiment. (2) The ocean uptake of carbon comes to dominate the carbon cycle as emissions decline, a feature that cannot be captured when forcing a model with the 1% experiment, as emissions always increase in that experiment. (3) The permafrost carbon feedback to climate change under the 1% experiment forcing is less than half the strength of the feedback seen under logistic experiment forcing. Using the logistic experiment also allows smooth transition to zero or negative emission states, allowing these states to be examined without sharp discontinuities in $CO_2$ emissions. The protocol for the CMIP6 iteration of $C^4MIP$ again sets the 1% experiment as the benchmark experiment for model intercomparison, however clever use of the Tier 2 experiments may alleviate some of the limitations outlined here. Given the limitations of the 1% experiment as the benchmark experiment for carbon cycle intercomparisons, adding a logistic or similar idealized experiment to the protocol of the CMIP7 iteration of $C^4MIP$ is recommended.

## 1   Introduction

Idealized climate change experiments are used as common framework to compare the output of ensembles of climate models (Houghton et al., 1996). These experiments are used to estimate standard Earth system metrics such as: Climate Sensitivity (Gregory et al., 2004), Transient Climate Response (Houghton et al., 2001; Raper et al., 2002), and Transient Climate Response to Cumulative $CO_2$ Emissions (TCRE) (Gillett et al., 2013). Two of the idealized experiments outlined by the Climate Model Intercomparison Project (CMIP) prescribe changes only in atmospheric $CO_2$ concentration, the $4\times CO_2$ experiment and the 1% per year compounded increased in atmospheric $CO_2$ experiment (hereafter referred to as the 1% experiment) (Meehl et al., 2007; Taylor et al., 2012; Eyring et al., 2016). Using either of these experiments allows for the study of the effect of $CO_2$ on

the Earth system without having to account for the confounding effects of land-use-change, non-$CO_2$ greenhouse gases, and aerosols. The $4\times CO_2$ experiment prescribes an instantaneous quadrupling of atmospheric $CO_2$ concentration (relative to pre-industrial concentration) and is principally used to estimate equilibrium climate sensitivity (Gregory et al., 2004; Rugenstein et al., 2016). The 1% experiment prescribes a rise in atmospheric $CO_2$ concentration from pre-industrial concentration at 1% a year compounded, resulting in a doubling of $CO_2$ concentration at year 70 of the simulation, and a quadrupling of $CO_2$ concentration at year 140 of the simulation. The 1% experiment has principally been used to derive the Transient Climate Response, defined as the temperature change at the time of atmospheric $CO_2$ doubling in a 1% experiment (Houghton et al., 2001; Collins et al., 2013). As part of CMIP phase 5 (CMIP5) the 1% experiment was used to compare the carbon cycle feedbacks within Earth System Models (ESMs) (Taylor et al., 2012; Arora et al., 2013), and to derive values of TCRE and thus to compute carbon budgets compatible with various temperature targets (Gillett et al., 2013). A similar modelling protocol has been established for forthcoming CMIP6 simulations (Jones et al., 2016b). Here we critically examine this choice and propose a more suitable idealized experiment for examining carbon cycle feedbacks to climate change.

The first documented use of the 1% experiment was by Stouffer et al. (1989) where the experiment was used to force an early ocean-atmosphere general circulation model developed by the Geophysical Fluid Dynamics Laboratory [Personal Communication, R.J. Stouffer]. The 1% rate of increase was chosen to approximate the rate of increase of all anthropogenic greenhouse gasses observed at the time Stouffer et al. (1989) (while not accounting for cooling from aerosols, a process that was poorly quantified in the 1980s (Houghton et al., 2001)). The exponential functional form of the experiment was chosen because radiative forcing from $CO_2$ is approximately a logarithmic function of change in atmospheric $CO_2$ concentration. Therefore, an exponential rise in atmospheric $CO_2$ at 1% a year compounded results in a 1% a year linear increase in radiative forcing at the top of the atmosphere (Stouffer et al., 1989). Thus the increase in $CO_2$ was intended to represent the rise in all greenhouse gases. It was well understood that $CO_2$ concentration rose much faster in the 1% experiment than $CO_2$ concentration was expected to rise in the natural world (Stouffer et al., 1989). Since yearly records began in 1958 the annual rate of change in atmospheric $CO_2$ concentration has ranged between 0.09% in 1964 and 0.8% in 1998 with a mean of 0.43% from 1958 to 2016 (Figure 1). The numerical experiment of Stouffer et al. (1989) was included in the First Assessment Report of the IPCC Houghton et al. (1992) and by the time of the Second assessment report the 1% experiment had become a standard benchmark numerical experiment for climate model intercomparison (Houghton et al., 1996). In preparation for CMIP6 the 1% experiment has been incorporated into the CMIP Diagnostic, Evaluation and Characterization of Klima (DECK) protocol, intended to be the CMIP core set of experiments for the indefinite future (Eyring et al., 2016).

The development of Earth System Model (climate models including carbon cycles (Planton, 2013)) created a need for a transient $CO_2$-only experiment for use in model lncomparisons of this new class of model (Friedlingstein et al., 2006). In the first intercomparison of ESMs, a modified version of the Special Report on Emissions Scenarios A2 experiment was used, where non-$CO_2$ forcing and land-use-changes from the scenario were turned off for the simulation (Friedlingstein et al., 2006). However, later studies utilizing model output from the Coupled Climate–Carbon Cycle Model Intercomparison Project ($C^4MIP$) implicitly criticized the choice of the A2 scenario (Gregory et al., 2009; Matthews et al., 2009). Gregory et al. (2009) recommended using the 1% experiment in place of a modified scenarios, due to the simplicity of the 1% experiment, the

experiment's well established role in model intercomparison projects, and the magnitude of emissions implied by the 1% experiment being of similar magnitude to socioeconomic scenarios. This recommendation was implemented, with the CMIP5 protocols calling for benchmark carbon cycle experiments to be carried out using a 1% experiment (Taylor et al., 2012). The protocol for the CMIP6 iteration of C[4]MIP also calls for the 1% experiment to be used at the benchmark for carbon cycle model intercomparison, along with a selection of scenario based simulations (Jones et al., 2016b).

The carbon cycle is classically subdivided into the terrestrial carbon cycle, dominated by plant and soil biology, and the oceanic carbon cycle, dominated by ocean carbonate chemistry but influenced by ocean biogeochemistry (Ciais et al., 2013). Together these processes remove over half of the carbon emitted to the atmosphere by burning of fossil fuels and land-use-change, greatly mitigating the effect of $CO_2$ emissions on climate change (e.g. Le Quéré et al., 2018). In the terrestrial domain there are three major feedbacks that are induced by increases in atmospheric $CO_2$ concentration and resultant climate warming. The response of plants in $CO_2$ limited ecosystems to increased atmospheric concentration of $CO_2$ is the $CO_2$ fertilization effect, whereby primary productivity is increased leading to larger plant biomass and an increased flux of dead biomass into soils (e.g. Ciais et al., 2013). The rate of heterotrophic respiration of soil organic matter is strongly controlled by temperature (Jenkinson et al., 1991), such that warmer soil temperature tend to induce a faster overturn of soil organic matter and release of carbon to the atmosphere (Jenkinson et al., 1991). This enhanced soil respiration is especially significant in permafrost environments, where the switch from dominantly frozen soils to dominantly thawed soils dramatically increases the rate of respiration and can lead to the release of long-sequestered pools of carbon (Schuur et al., 2015). The third major feedback affecting the terrestrial realm is ecosystem changes resulting from climate warming and can be either a positive or negative feedback depending on the region of the world (e.g. Malhi et al., 2009; Ciais et al., 2013).

In the ocean domain uptake of carbon is driven by the difference in the partial pressure of $CO_2$ ($pCO_2$) between the surface ocean and the atmosphere (e.g. Greenblatt and Sarmiento, 2004). Where atmospheric $CO_2$ concentration is higher than local sea surface $pCO_2$, atmospheric $CO_2$ will invade the ocean system. Due to ocean carbonate chemistry most of the carbon that enters the ocean is reacted to bicarbonate ions, with a relatively small fraction held as dissolved $CO_2$ – the species that controls ocean $pCO_2$ (e.g. Greenblatt and Sarmiento, 2004). Like most gases dissolved $CO_2$ has a higher partial pressure in warm water relative to colder water such that climate warming is expected to reduce the efficiency of ocean carbon uptake (Ciais et al., 2013). On centennial timescales the ocean has a relatively fixed alkalinity, such that the ocean carbonate chemistry equilibrium will shift to a state where more carbon is held as dissolved $CO_2$ – an effect which also reduces the efficiency of ocean carbon uptake (Broecker and Peng, 1982). Over longer time periods the ocean will increase its alkalinity by dissolving calcium carbonate from ocean sediments, allowing the ocean to absorb additional $CO_2$ from the atmosphere (Archer, 1996). Thus, the ocean is expected to become less efficient at absorbing carbon as $CO_2$ emissions continue but is expected to continue to be a carbon sink far into the future (Arora et al., 2013). However, feedbacks involving ocean biogeochemistry and changes in overturning circulation remain important but poorly quantified uncertainties (Jones et al., 2016b).

Carbon cycle feedbacks are affected both by the concentration of atmospheric $CO_2$, air temperature, and the rates of change of these qualities (e.g. Greenblatt and Sarmiento, 2004). A faster rise in atmospheric $CO_2$ will increase the partial pressure gradient between the atmosphere and the ocean, tending to increase the rate of ocean carbon invasion, however the ocean will

have less time to overturn the mixed layer tending to reduce the invasion rate. The $CO_2$ fertilization effect depends on $CO_2$ concentration but the buildup of carbon in soils from higher primary productivity and thus enhanced soil respiration depends on the rate of change of $CO_2$. Permafrost carbon feedbacks in particular are sensitive to the rate of change in temperature as it takes time to thaw soils and decay the highly recalcitrant permafrost carbon pool (e.g. Schuur et al., 2015). Therefore, using the 1% experiment as the benchmark for carbon cycle model intercomparison may be problematic given that $CO_2$ rises in the experiment at twice the historical rate. Such concerns are supported by a recent study utilizing an intermediate complexity ESM with a representation of the permafrost system, and forced with the 1% experiment. The study showed a permafrost carbon feedback strength evaluated at year 70 of the 1% experiment of only $\sim 8\%$ of the feedback strength evaluated in the year 2100 CE of the RCP 8.5 scenario with the same model (MacDougall et al., 2017; MacDougall and Knutti, 2016). MacDougall et al. (2017) concluded that the 1% experiment warmed too fast to allow permafrost to thaw. The study recommended a different idealized $CO_2$-only experiment be developed to help evaluate effect of permafrost carbon feedbacks on climate change. Here we develop such a new idealized scenario and compare it to the 1% experiment.

## 2   Methods

### 2.1   Model Description

The University of Victoria Earth System Climate Model (UVic ESCM) is a climate model of intermediate complexity that participate in both the original $C^4MIP$ and the CMIP5 iteration of $C^4MIP$ (Friedlingstein et al., 2006; Arora et al., 2013). The core of the model is a full three dimensional ocean general circulation model coupled to a simplified moisture and energy balance atmosphere (Weaver et al., 2001). The UVic ESCM contains a detailed representation the global carbon cycle including oceanic and terrestrial components. The ocean carbonate chemistry is simulated following the protocols of the ocean carbon cycle model intercomparison project (Orr et al., 1999), and ocean biogeochemistry is represented using a nutrient-phytoplankton-zooplankton-detritus ocean biology scheme (Schmittner et al., 2008). The slow dissolution of ocean carbonate sediments follows (Archer, 1996). The terrestrial carbon cycle is represented using the Top-down Representation of Interactive Foliage and Flora Including Dynamics (TRIFFID) dynamic vegetation model (Meissner et al., 2003; Matthews et al., 2004; Cox et al., 2001). The version of the model used in the present study is a modified variant of the frozen ground version of the UVic ESCM which includes a representation of the permafrost carbon pool (MacDougall and Knutti, 2016), and is the same version of the model used in MacDougall et al. (2017) where the limitations of the 1% experiment with respect to the permafrost carbon feedback were first encountered.

### 2.2   A new idealized experiment

We can conceptualize $CO_2$ emission pathways as having four potential stages: (1) increasing emissions, that are captured by the 1% experiment and also by the modified A2 scenario used by the original $C^4MIP$ (Friedlingstein et al., 2006). (2) Decreasing emissions, the stage of emissions following peak emissions captured by many climate scenarios (Meinshausen et al., 2011).

(3) Zero emissions, a stage used to investigate the behaviour of the carbon cycle after $CO_2$ emissions cease (e.g. Matthews and Weaver, 2010; Frölicher et al., 2014). And (4) negative emissions, a stage used to investigate the behaviour of the carbon cycle during a hypothesized mass deployment of artificial atmospheric $CO_2$ removal technology (e.g. Samanta et al., 2010; Boucher et al., 2012; Zickfeld et al., 2013). Previous studies have used the multi-gas RCP 2.6 scenario to examine increasing, decreasing, and negative emission stages (Jones et al., 2016a). Some studies have used a fifth stage of the emissions pathway where the emission rate is constant (e.g. Krasting et al., 2014). Although useful for some examining some problems, such a state not necessary to capture the likely evolution of $CO_2$ emissions. Thus an ideal idealized scenario should include both an increasing and decreasing phase, and allow for a smooth transition to zero emissions or negative emission potentialities.

There are an infinite number of $CO_2$ pathways that could satisfy these core criteria. Thus two more constraints are added: that the pathway roughly follow the $CO_2$ trajectory of the historical record (Trans and Keeling, 2017), and the the pathway be an elementary function. Given these criteria a logistic function was settled upon:

$$C_A = \frac{C_{Ap}}{1 + e^{-k(t-t_m)}} + C_{Ao} \tag{1}$$

where $C_A$ is the atmospheric $CO_2$ concentration, $C_{Ao}$ is the original atmospheric $CO_2$ concentration, $C_{Ap}$ is peak atmospheric $CO_2$ concentration, $t$ is time, $t_m$ is the mid-point of the function, and $k$ is a rate constant. $k$ and $t_m$ are found by fitting the function to the historical $CO_2$ record. $C_{Ao}$ is taken to be 280 ppm and $k$ and $t_m$ values for $2\times$, $4\times$, and $8\times$ pathways are shown in Table 1, these pathways are displayed in Figure 2a. Logistic functions are common in nature and appear in systems where growth is limited by finite resources (e.g Reed and Berkson, 1929). Famously a logistic equation is one solution to the Verhulst-Lotka population growth equations (e.g Berryman, 1992), and RCP extension scenarios (except for the peak-and-decline RCP 2.6) resemble logistic functions (Moss et al., 2010).

The $4\times$ $CO_2$ logistic pathway is compared to the historical $CO_2$ trajectory, the $4\times$ $CO_2$ 1% experiment pathway, and the pathway of Wigley and Schlesinger (1985) in Figure 3. The Wigley and Schlesinger (1985) pathway was an early alternative to the 1% experiment the fell out of use before IPCC SAR. The figure demonstrates how fast $CO_2$ concentration grows in the 1% experiment relative to to the historical record and also how much closer the Wigley and Schlesinger (1985) pathway trends to the post 1985 historical record. Logistic functions asymptote towards their high and low bounds as the function approached positive and negative infinity, hence how close the function is to the peak and preindustrial $CO_2$ concentration depends on the match between the logistic function and the historical $CO_2$ record. Treating year 1850 as the preindustrial reference time and making the function symmetric about its mid-point leads to preindustrial and peak $CO_2$ values within 1 ppm of the target value for the three derived pathways.

## 2.3 Model experiments

The UVic ESCM was forced with three versions of the 1% experiment and three logistic $CO_2$ pathways, with $CO_2$ concentrations reaching a peak of $2\times$, $4\times$, and $8\times$ pre-industrial concentration. Following peak $CO_2$ concentration model experiments branch to a zero emissions scenario and a negative emissions scenario. In the zero emissions scenario the model is switched

from prescribed atmospheric $CO_2$ to freely evolving $CO_2$ with fossil fuel emission rate set to zero, such that natural sources and sinks determine atmospheric $CO_2$ concentration (Eby et al., 2009). In the negative emissions scenario $CO_2$ concentration return to pre-industrial concentration in a mirror image of their original rise (Figure 2). The mirrored return negative-emissions-scenario derived from the 1% experiment is the 1%–up 1%–down experiment used by several previous studies (Samanta et al., 2010; Boucher et al., 2012; Zickfeld et al., 2013, 2016; Schwinger and Tjiputra, 2018). The 1%–up 1%–down experiment has been incorporated as a standard model experiment for CMIP6 as part of the Carbon Dioxide Removal (CDR) MIP (Keller et al., 2018). In model simulations where atmospheric $CO_2$ concentration is prescribed, anthropogenic emissions are diagnosed from conservation of mass as the residual of the carbon cycle.

The UVic ESCM is being used here simply to illustrate the differences between the behaviour of the carbon cycle in the 1% and logistic experiments. Replication of the logistic experiment with different ESMs is necessary to confirm the result presented below. In particular the behaviour of the terrestrial carbon pool is likely to be different in different ESMs Friedlingstein et al. (2006); Arora et al. (2013).

## 3   Results

### 3.1   Increasing & Decreasing Emissions

The evolution of atmospheric $CO_2$ concentration under the logistic and 1% experiments are shown in Figure 2. The figure also shows the diagnosed emissions for each of the pathways. The emissions for the logistic experiment grow slowly for the first 100 years of the simulation, then enter a stage of rapid increase before reaching peak emissions and going into a decreasing phase. After the peak, emissions decline rapidly before entering a long tail of low but persistent emissions. For the 1% experiment emissions monotonically increase through time, with the rate of change in emissions slowing as $2\times$ $CO_2$ is approached but increasing again as $4\times$ $CO_2$ is approached. For the logistic experiments $CO_2$ emissions peak at 10, 17 and 28 $\mathrm{Pg\,C\,a^{-1}}$ respectively for the $2\times$, $4\times$, and $8\times$ pathways. For the 1% experiments emissions always peak at the termination of the experiment with values of 21, 29 and 48 $\mathrm{Pg\,C\,a^{-1}}$ respectively for the $2\times$, $4\times$, and $8\times$ pathways. For perspective recall that 2017 anthropogenic $CO_2$ emissions are 11 $\mathrm{Pg\,C\,a^{-1}}$ (Le Quéré et al., 2018).

A key purpose of carbon cycle models and related model intercomparison projects is to explore how the ocean system and land biosphere carbon sinks will operate under changed climate conditions (e.g. Friedlingstein et al., 2006). One way of visualizing these processes is through the fraction of emitted carbon that is stored in each of the main fast-cycling carbon reservoirs. Hence we can define airborne, ocean-borne and land-borne fractions of carbon. These can be defined either as instantaneous fractions, the fraction of carbon emitted this year that ends up in each reservoir, or as cumulative fractions – the fraction of carbon emitted since pre-industrial times held in the ocean, land, and atmosphere. Instant fractions give an immediate sense of how the Earth system is reacting to changes in $CO_2$ concentration and temperature but are not defined once emissions reach zero. Figure 4 shows the instantaneous and cumulative airborne, ocean-borne, and land-borne fractions of carbon for the logistic and 1% $4\times$ $CO_2$ experiments. The increasing phase of the logistic experiment closely resembles the 1% experiment, with closely matched ocean-borne and land-borne fractions early in the simulation, a gradual rise in the airborne

fraction of carbon and decline in the land-borne fraction near the end of the increasing phase. In the declining emission phase of the logistic experiment the carbon cycle behaves differently. As emissions decline, the land system transitions from a carbon sink to a carbon source. The ocean sink comes to dominate the system absorbing more carbon than the anthropogenic $CO_2$ emissions to the atmosphere.

Figure 6 shows the evolution of the land carbon pool anomalies in the $4\times CO_2$ 1% and logistic experiments. The land pool is shown broken down into carbon held in living vegetation, soil carbon, and permafrost carbon (carbon that had been frozen in permafrost soil layers at the beginning of the simulation, which maintains distinct properties after being thawed in the UVic ESCM model scheme (MacDougall and Knutti, 2016)). The figure shows that in both experiments the permafrost carbon pool declines monotonically, and vegetation carbon increases with $CO_2$ concentration. However, in the logistic simulation the soil

carbon pool increases to a peak anomaly of 259 Pg C before declining to near zero by the end of the simulation. Under the 1% experiment the soil carbon pool anomaly peaks but only just begins a decline before the simulation ends. Hence an enhanced soil respiration feedback is evident in the logistic experiment but not present in the 1% experiment.

   Figure 5 displays the change in vegetation and soil carbon between pre-industrial conditions and the time of doubled atmospheric $CO_2$ under the $4\times CO_2$ 1% and logistic experiments. The spatial patterns of change are similar under both experiments,

but of greater magnitude under the logistic experiment. Vegetation experiences a loss of carbon in the Andes and in mid-latitude northern extra-tropics, while gains in vegetation carbon are seen in the topics, subtropics, sub-arctic and arctic regions. Soils show a reduction in carbon in the permafrost region, boreal forests, and Sahel. Increases in soil carbon in seen in central North America, central Eurasia, and southern Africa, regions generally corresponding to grasslands. Overall the figure shows complex biome-specific responses of the terrestrial biosphere to increasing atmospheric $CO_2$ concentration.

The second key feature of the carbon cycle under declining emissions is the increase in the instantaneous ocean-borne fraction of carbon (Figure 4b). The feature is explored in Figure 7 which shows the absolute uptake of carbon by the land, ocean, and atmosphere. The figure shows that ocean carbon uptake does decline in the logistic experiment as the $CO_2$ emission rate declines, however the ocean remains a sink and by the end of the simulation is absorbing carbon at 1.3 Pg C a$^{-1}$. Given that in multi-millennial ESM simulations the ocean tends to absorb carbon for many centuries after emissions cease (e.g. Eby et al.,

2009; Randerson et al., 2015) this feature is not unexpected, and is evident for model simulations under the peak-and-decline RCP 2.6 scenario in CMIP5 ESM output (Jones et al., 2013).

   The motivation to create the logistic pathway experiments was the weak response on the permafrost carbon feedback under the 1% experiment relative to the feedback strength seen under the RCP scenarios (MacDougall et al., 2017). The response of carbon in the permafrost region (including both the modelled permafrost carbon pool and regular soil carbon in the active

layer above the permafrost) is shown in Figure 8. At the mid-point of the $4\times CO_2$ simulations, where carbon cycle feedbacks are classically evaluated (e.g. Arora et al., 2013), the permafrost region has released 19 PgC under the 1% experiment and 55 PgC under the logistic experiment. By the end of each simulations the permafrost region releases 99 PgC under the 1% experiment and 238 PgC under the logistic experiment. Figure 8 demonstrates the importance of elapsed time in destabilizing permafrost carbon. The logistic experiment implies lower $CO_2$ emission rate and hence a lower rate of warming, results in a

higher release of carbon from permafrost regions at any given $CO_2$ concentration. The result is consistent with previous work

on the permafrost carbon feedback, which demonstrates a long lag time between forcing and response due to the time taken to thaw soil and decay soil carbon (e.g. Schuur et al., 2015).

Cumulative emissions versus temperature change curves and TCRE values for the $2\times$, $4\times$, and $8\times$, 1% and logistic experiment simulations are shown in Figure 9. The TCRE relationship in general shows strong independence from forcing scenario, (e.g. MacDougall et al., 2017), a feature which is evident in Figure 9. Near the end of the logistic experiments when the rate of implied $CO_2$ emissions slows, the TCRE values deviate from scenario independence. Theoretical work on the TCRE relationship suggests that path independence should break-down at very high and very low emission rates (MacDougall, 2017). The results shown in Figure 9 and consistent with this understanding. At the time of $CO_2$ doubling the simulated TCRE value is 1.6 EgC $K^{-1}$ under all experiments except the $2\times$ logistic experiment where that value is 1.7 EgC $K^{-1}$.

## 3.2 Zero Emissions

An important question posed to ESMs concerns evolution of atmospheric $CO_2$ concentration and global temperature following total cessation of net $CO_2$ emissions (e.g. Matthews and Weaver, 2010; Frölicher et al., 2014). Whether global temperature will increase, decrease, or stabilizes following cessation of emissions determines the final size of carbon budgets compatible with temperature targets, and has important policy implications (e.g. Frölicher et al., 2014). The change in temperature following cessation of $CO_2$ emissions is termed the Zero Emissions Commitment (ZEC) (Zickfeld et al., 2012). The evolution of atmosphere $CO_2$ and cumulative carbon fractions following cessation of emissions for the logistic and 1% experiments is shown in Figure 10. In both experiments atmospheric $CO_2$ concentration drops after emissions cease, consistent with most other ESMs (Frölicher et al., 2014). The carbon fractions for both experiments are similar after emissions cease, with the airborne and land-borne fractions of carbon declining and the ocean-borne fraction of carbon increasing. The difference between the logistic and 1% experiments is most evident at the time emission stop. The logistic experiment has a continuous behaviour as emissions increase, decrease then reach zero, while the 1% experiment exhibits a sharp discontinuity in behaviour when emissions cease (Figure 10). The airborne fraction goes from increasing to decreasing, the ocean-borne fraction goes from stable to increasing, and the land-borne fraction goes from declining quickly to declining slowly. Figure 11 shows the evolution of the surface air temperature anomaly for the logistic and 1% experiments during the transient simulation and following cessation of emissions. Both experiments have positive ZEC, with the logistic experiment warming 0.6°C between cessation of emissions and year 1000 of the experiment, and the 1% experiment warming 1.5°C between cessation of emissions an year 1000. Despite having a smaller ZEC the logistic experiment is warmer than the 1% experiment in the year 1000 by 0.3°C.

Figure 11b shows the radiative forcing and ocean heat uptake under both the 1% and logistic ZEC experiments. The figure shows that under the 1% experiment radiative forcing and ocean heat uptake peak the moment emissions cease. While under the logistic experiment ocean heat uptake peaks over a century before emission cease. The declining ocean heat uptake under that logistic experiment explains the smaller ZEC under that experiment. When emissions cease under the logistic experiment the Earth system is closer thermal equilibrium resulting in a smaller radiative imbalance and unrealized warming. These results are consistent with previous experiments examining the pathway dependence of ZEC (Ehlert and Zickfeld, 2017).

### 3.3 Negative Emissions

Recent interest in the possibility of net negative $CO_2$ emissions being used to undo climate change through 'carbon remedi-ation' (e.g. Shepherd, 2009) has lead to the formalization of the 1%–up 1%–down idealized experiment. In this experiment atmospheric $CO_2$ concentration follows the 1% experiment up to a given threshold, then is returned to preindustrial concen-tration in a mirrored path with -1% compounded reductions in $CO_2$ concentration each year (Samanta et al., 2010; Boucher et al., 2012; Zickfeld et al., 2013, 2016; Schwinger and Tjiputra, 2018). Simulations of the 1%–up 1%–down experiment and the logistic equivalent for 2×, 4×, and 8× $CO_2$ are shown in Figure 2. This figure also shows $CO_2$ emissions diagnosed from these pathways. The logistic pathways exhibit a smooth transition to negative emissions, with declining rate of emissions sim-ply continuing to decline once the zero emissions threshold is breached. The 1%–up 1%–down pathways create a transition from very high positive emissions to very large negative emissions over the course of a single year. For example, under the 4× $CO_2$ 1% up 1% down emissions go from 29 PgC $a^{-1}$ in year 140 of the experiment to -21PgC $a^{-1}$ in year 142 of the experiment. The abrupt change in emission rate also leads to an abrupt cooling of the climate as shown in Figure 12. The figure shows that the 1%–up 1%–down experiment exhibits a linear increase in temperature followed by a fast but asymmetric cooling of temperature with a long tail of warming which dies out by about the year 500 of for the 4× $CO_2$ simulation. The logistic mirrored experiment shows a far more asymmetric temperature curve, with temperature continuing to increase for over a century once $CO_2$ concentration begins to drop (consistent with the ZEC results) followed by a slow decline in temperatures, with temperature still 0.85 °C above preindustrial by the end of the simulation in model year 1200.

## 4 Discussion

A critical question from the early days of coupled carbon cycle modelling was whether the terrestrial carbon sink will transition to source of carbon to the atmosphere (e.g. Cox et al., 2001). One model in each of the two previous intercomparisons of the carbon cycles of ESMs exhibited a terrestrial sink to source transition, HadCM3LC in Friedlingstein et al. (2006), and MIROC-ESM in Arora et al. (2013). However, as all the other models in those intercomparisons did not exhibit a terrestrial sink to source transition this potentiality would appear possible but unlikely (Friedlingstein et al., 2006; Arora et al., 2013). Here we have shown that at least to some degree that the existence of a terrestrial sink to source transition is contingent on the choice of idealized experiment used to force a model. Had a logistic experiment been used in previous iterations of C[4]MIP different conclusions about the robustness of the terrestrial carbon sink may have been drawn.

In order to meet the 2°C temperature target outlined in the Paris Agreement (United Nations, 2015) emissions of $CO_2$ should peak during the present decade (Rogelj et al., 2011; Friedlingstein et al., 2014). Even under less aggressive temperature targets emissions rate should peak in the early to mid 21st century (Rogelj et al., 2011; Friedlingstein et al., 2014). Thus examining the behaviour of the carbon cycle under conditions of declining emissions is a area of imminent importance (e.g. Jones, 2017). Here we have shown that the ocean comes to dominate the global carbon cycle as emissions decline. This feature of the carbon cycle has actually appeared in previous intercomparisons of intermediate complexity ESMs forced with multi-gas scenarios (Zickfeld et al., 2013). However, ocean dominance has not been emphasized in literature. For example the summary for policy

makers for the IPCC AR5 states that: "Based on Earth System Models, there is high confidence that the feedback between climate and the carbon cycle is positive in the 21st century; that is, climate change will partially offset increases in land and ocean carbon sinks caused by rising atmospheric $CO_2$. As a result more of the emitted anthropogenic $CO_2$ will remain in the atmosphere." (IPCC, 2013). Although this statement is strictly true, carbon sinks are weaker in fully coupled models relative

to biogeochemically coupled models (Arora et al., 2013), the statement does not make clear that if emissions rate peaks then declines, a smaller fraction of emitted anthropogenic carbon will remain in the atmosphere. The more extensive discussion of carbon cycle feedbacks in Chapter 6 of AR5, also makes no mention of a higher ocean-borne fraction under declining emission rates Ciais et al. (2013). Thus examining the ocean carbon cycle as emissions decrease should be a key priority when analyzing the model output of the CMIP6 iteration of $C^4MIP$.

A key priority of the CMIP6 iteration of $C^4MIP$ is examining the strength of the permafrost carbon cycle feedback to climate change (Jones et al., 2016b), as representations of permafrost carbon were absent from the models that participated in CMIP5 (Arora et al., 2013). Informal intercomparisons of the permafrost carbon feedback have thus far used year 2100 of RCP 8.5 as the point of comparison (Schuur et al., 2015). Given the results of the present study and MacDougall et al. (2017) changing the point of comparison to year 70 of the 1% experiment is likely to substantially underestimate the potential contribution

of permafrost carbon to climate change. Thus the point of comparison and the lagged nature of the permafrost carbon cycle feedback must be explicitly acknowledged when analysis of the CMIP6 iteration of $C^4MIP$ is conducted.

Zero emissions scenarios and full mirroring of atmospheric $CO_2$ back to pre-industrial concentration are experiments outside the realm of $C^4MIP$ (Jones et al., 2016b), and generally investigated with Earth system models of intermediate complexity (e.g. Zickfeld et al., 2016) or with a single simulation with a full ESM (e.g. Zickfeld et al., 2012; Frölicher et al., 2014). Thus it

is relatively simple for investigators studying these post-emissions states to switch to a logistic like experiment or a scenario which includes declining emissions. That the choice of which idealized experiment to use can drastically change results of ZEC or negative emissions experiments should gravely concern investigators and reinforce the need for applying critical thought to experiment design. The 1%–up, 1%–down experiment in particular needs to be critically examined as an abrupt transition from very high positive emissions to very large negative emissions is deeply implausible. All idealized scenarios are implausible in

some aspects but generally similar events could occur as a component of broader cataclysms. An abrupt transition from high to zero emissions as seen in the 1% to zero emissions experiment is consistent with a nuclear war obliterating the infrastructure of the fossil fuel economy (Turco et al., 1983). Even the abrupt release of $CO_2$, doubling or quadrupling atmospheric $CO_2$ concentration is more plausible than it may first appear. Paleoclimate evidence from the end-Cretaceous impact event suggests an increase in atmospheric $CO_2$ concentration of about this magnitude, originating from vaporized rock and global firestorms

(MacLeod et al., 2018). However, there is no plausible way to decarbonize the global economy and deploy atmospheric $CO_2$ removal technology in a single year. Thus investigators considering using the 1%–up 1%–down experiment should carefully examine why they are conducting the experiment and whether some other $CO_2$ pathway would be more appropriate.

## 4.1 Recommendations for analysis of CMIP6 iteration of C$^4$MIP

The protocol for the CMIP6 iteration of C$^4$MIP is now set in stone (Jones et al., 2016b). The two Tier 1 experiments required of all participating modelling groups are the 1% experiment and SSP5-8.5 scenario (the successor to RCP8.5) up to year 2100 CE. Both of these experiments prescribe monotonically increasing atmospheric $CO_2$ concentration and thus have no declining emission phase. However, two of the Tier two experiments, SSP5-8.5-BGC to 2300 and SSP5-3.4-Overshoot-BGC have declining emission phases and SSP5-3.4-Overshoot-BGC implies negative emissions (Jones et al., 2016b). Although using complex multi-forcing scenarios to evaluate carbon cycle feedbacks is sub-ideal in many ways using the Tier 2 scenarios may allow a more complete evaluation of the carbon cycle than using only the 1% experiment. Therefore I recommend:

1. Evaluating the permafrost carbon cycle feedback to climate change using the SSP5-8.5 and SSP5-8.5-BGC scenarios, not at year 70 of the 1% experiment.

2. Examining the relative strength of ocean carbon uptake during the decreasing emission phases of SSP5-8.5-BGC to 2300 and SSP5-3.4-Overshoot-BGC

## 4.2 Recommendations for incorporation of the logistic experiment into CMIP7

The key advantages of the logistic experiment over the 1% experiment are that the logistic experiment captures the phase of declining emissions, and allows for a smoother transition to zero emissions or negative emissions scenarios. The principle disadvantages of switching to the logistic experiment are the much higher computation cost of the experiment, and the loss of historical continuity in experiment design. Therefore several options are available for incorporation of the logistic experiment into the CMIP7 iteration of C$^4$MIP, ranging from full replacement of the 1% experiment, to incorporation of the logistic experiment into the Tier 2 experiment recommendations.

Given the high computation cost of feedback separation, which necessitates running fully coupled, radiatively coupled, and biogeochemically coupled simulations (Gregory et al., 2009; Arora et al., 2013), it is prohibitively expensive for the logistic experiment to be used for this purpose. Thus I recommend that the logistic experiment be added to the Tier 1 set of experiments, in addition to the 1% experiment. With the logistic experiment used for examining carbon fraction under declining emissions, and evaluating the strength of the permafrost carbon cycle feedback. Either the $2\times$ $CO_2$ or $4\times$ $CO_2$ versions of the logistic experiment could be incorporated into the Tier 1 experiments, with the $2\times$ $CO_2$ version having the advantage of being 'policy relevant', and less computationally demanding.

## 5 Conclusions

Idealized scenarios have a long history in the climate modelling community having been used for benchmark experiments and as a means of model intercomparison since the Second Assessment Report of the IPCC Houghton et al. (1996). In the Fifth Assessment Report the 1% experiment was used to compare the behaviour of models which included a representation of the global carbon cycle Ciais et al. (2013); Arora et al. (2013) and the 1% experiment will again be at the core of CMIP6

analysis of these systems (Jones et al., 2016b). However, in the 1% experiment atmospheric $CO_2$ concentration rises far faster than the historical record and implies only monotonically increasing $CO_2$ emission rate. Therefore, the experiment does not facilitate examination of the carbon cycle under conditions of declining emission rates, nor does the experiment allow for smooth transitions to zero emissions or negative emission experiments. Using a logistic shaped $CO_2$ concentration pathway

allows for study of increasing, decreasing, zero emissions and negative emissions states, and also better matches the historical trajectory of atmospheric $CO_2$ concentration.

By comparing simulations under the 1% experiment to simulations forced with a logistic $CO_2$ pathway, leading to the same atmospheric $CO_2$ concentration, we find five key differences: (1) simulations forced with the logistic experiment have a terrestrial sink to source transition, while simulations forced with 1% experiment do not. (2) Forced with the logistic experiment

the simulated ocean uptake of carbon comes to dominate the global carbon cycle as emissions decline. (3) Permafrost soils release less than half the carbon at the point of $CO_2$ doubling when forced with the $4\times CO_2$ 1% experiment relative to the $4\times CO_2$ logistic experiment simulations. (4) Following cessation of $CO_2$ emissions, the zero emissions commitment is much larger in simulations following the 1% experiment than for simulations following a logistic experiment. (5) Simulations with the 1%–up 1%–down experiment exhibits a smaller warming tail than simulations with the equivalent mirrored logistic experiment.

These differences suggest that the outcomes of many numerical climate experiments conducted with Earth systems models are contingent on the choice of $CO_2$ pathway used to force the model. Overall, I recommend adding a logistic-like experiment to the protocol of the CMIP7 iteration of $C^4MIP$.

*Data availability.* Model output from the 1% and logistic experiments described here are available for download from http://wiracocha.stfx. ca:5000/sharing/pZYta8JrV

*Code and data availability.* Version 2.9 of the UVic ESCM is available for download from http://climate.uvic.ca/model/. The updated model code including the frozen ground version of the model can be downloaded from ftp://uvicgroup:MalahatPa$$@data.iac.ethz.ch. Forcing files containing the $CO_2$ trajectories used for the numerical experiments described here are available in the supplementary data. Output variables necessary to reproduce the above numerical experiments are identical to those described in the $C^4MIP$ protocol. Refer to Section 4 of Jones et al. (2016b) for detailed description.

*Competing interests.* Author declares no competing interests.

*Acknowledgements.* This research was supported by a grant from the NSERC Discovery Grant programme. I thank Compute Canada for super-computer resources and support. R.J. Stouffer graciously provided an account of the of the origin and original purpose of the 1% experiment. I thank C.D. Jones and A.-I. Partanen for their thoughtful and critical reviews of this manuscript.

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

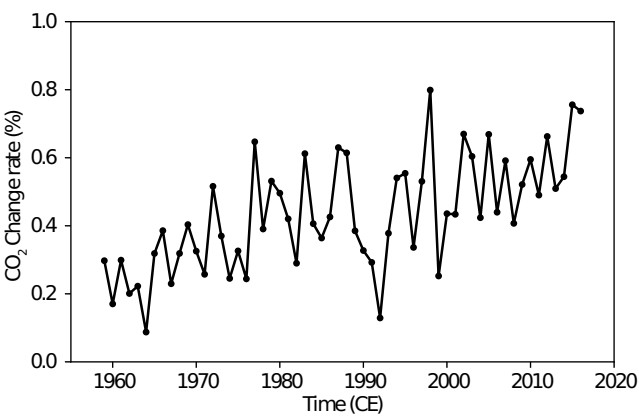

**Figure 1.** Historical yearly rate of atmospheric CO₂ change derived from the Mauna Loa record Trans and Keeling (2017). Note that the annual rate of increase has never breached 1% a year, and has averaged 0.46 % a$^{-1}$ over the historical record.

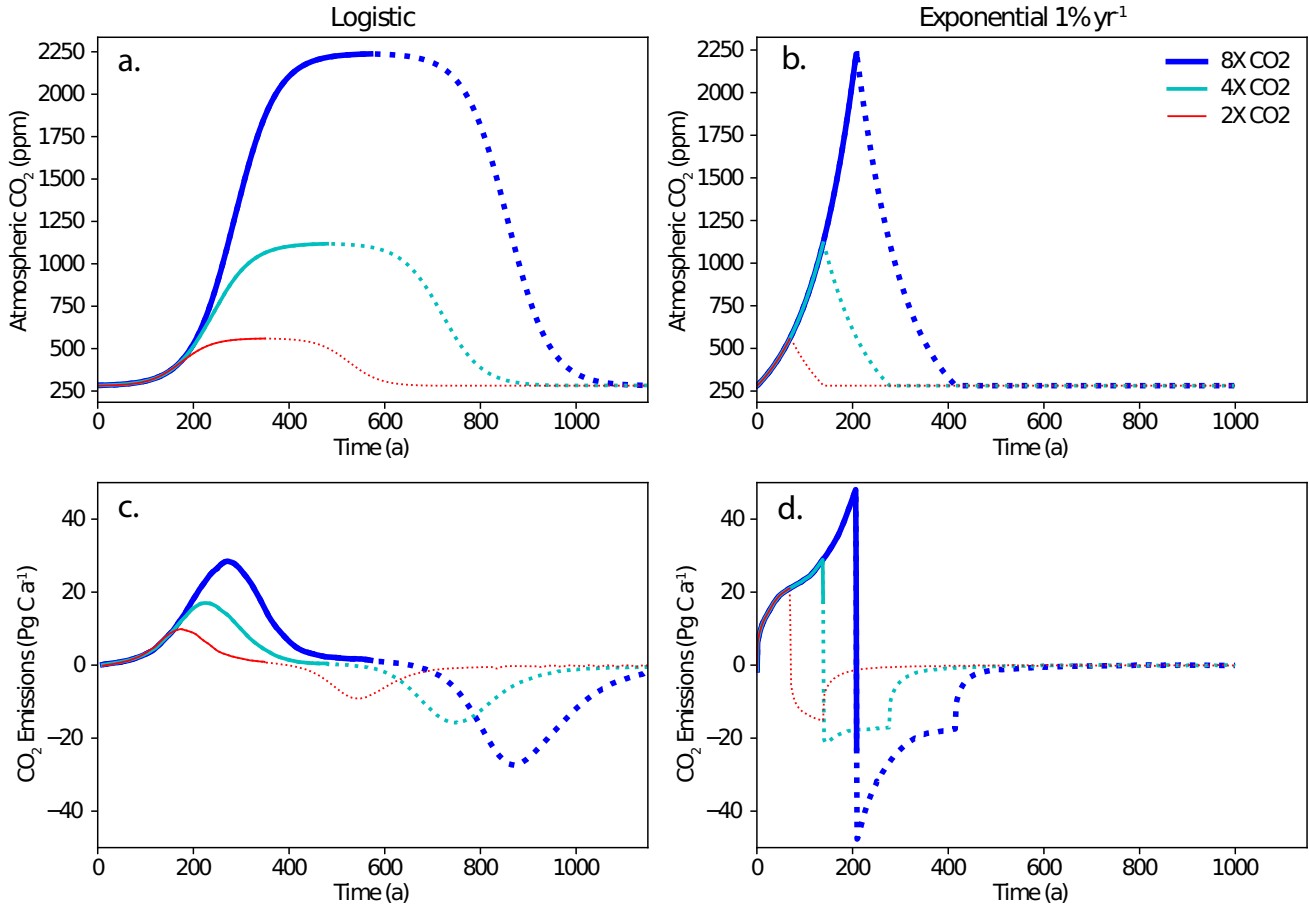

**Figure 2.** $CO_2$ pathways and diagnosed emissions for logistic and 1% experiment $2\times$, $4\times$, and $8\times$ $CO_2$ mirrored pathways. Solid lines are the increasing $CO_2$ phase of the simulation and dotted lines are the atmospheric $CO_2$ removal phase of the pathways. Note that the logistic $CO_2$ pathways extend over a much longer period of time than the 1% experiments. Also note that the diagnosed emissions are smooth under the mirrored logistic pathway, while emissions undergo a sharp discontinuity under the 1%–up 1%–down experiment.

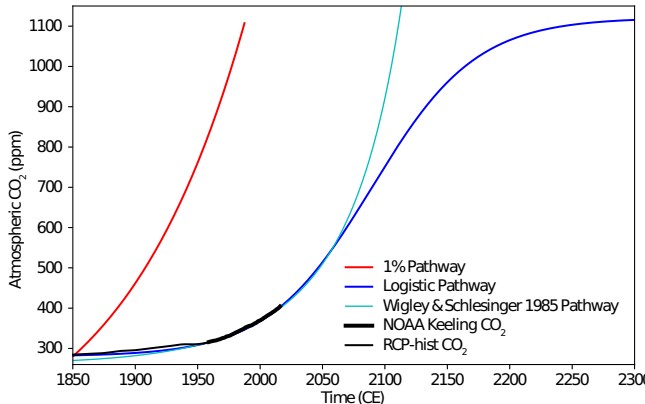

**Figure 3.** Historical atmospheric $CO_2$ trajectory and three idealized $CO_2$ pathways. Historical trajectories are from CMIP5 RCP-Historical archive for 1850 to 1958 (Moss et al., 2010) and from the National Oceanic & Atmospheric Administration (NOAA) record from Mauna Loa for 1959 to 2017 (Trans and Keeling, 2017). In addition to the $4\times CO_2$, 1% and logistic pathways the pathway of Wigley and Schlesinger (1985) (an abandoned alternative to the 1% experiment) is shown.

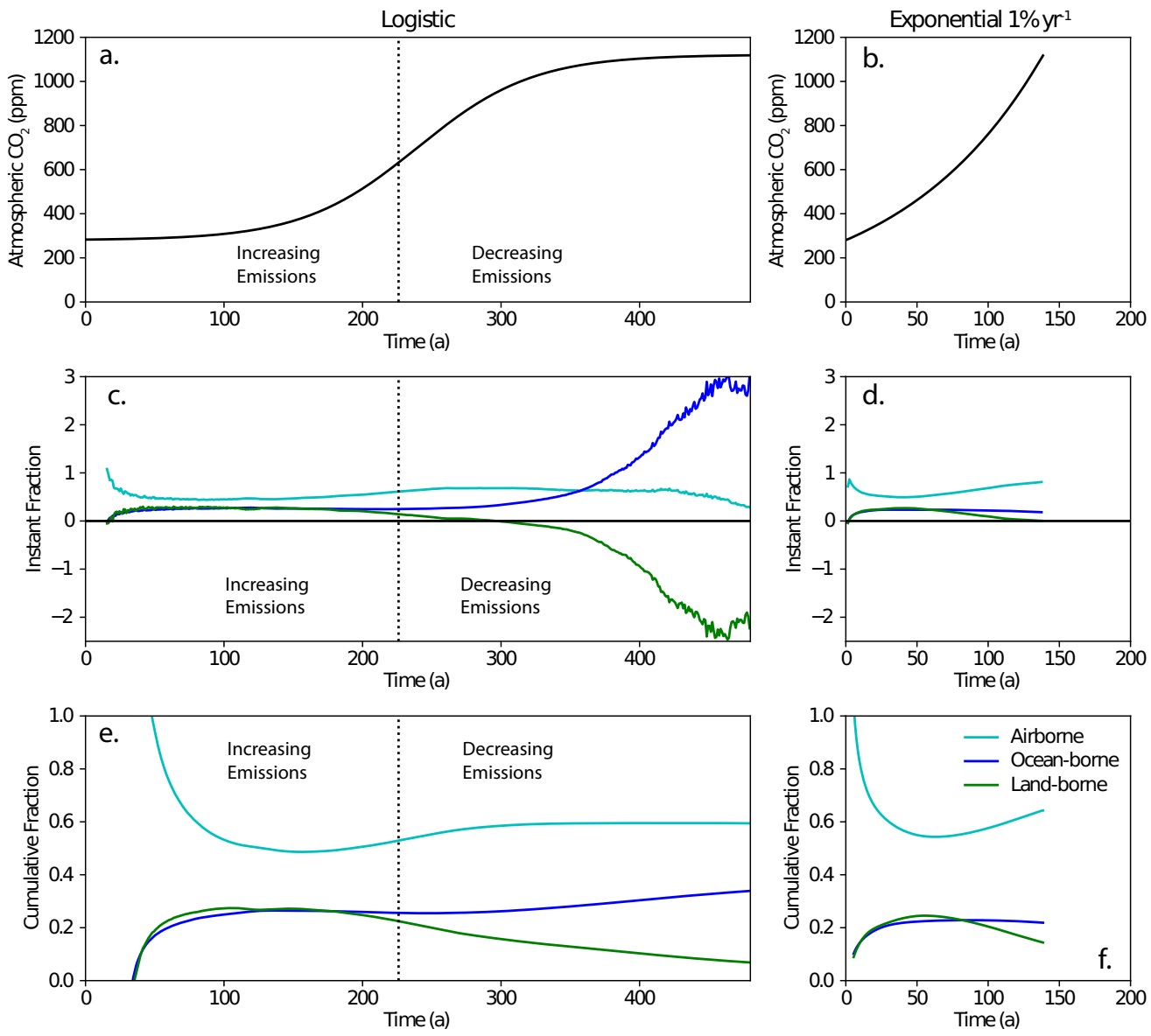

**Figure 4.** $CO_2$ concentration (a & b), instantaneous carbon fractions (c & d), and cumulative carbon fractions (e & f) for the $4\times CO_2$ logistic and 1% experiments.

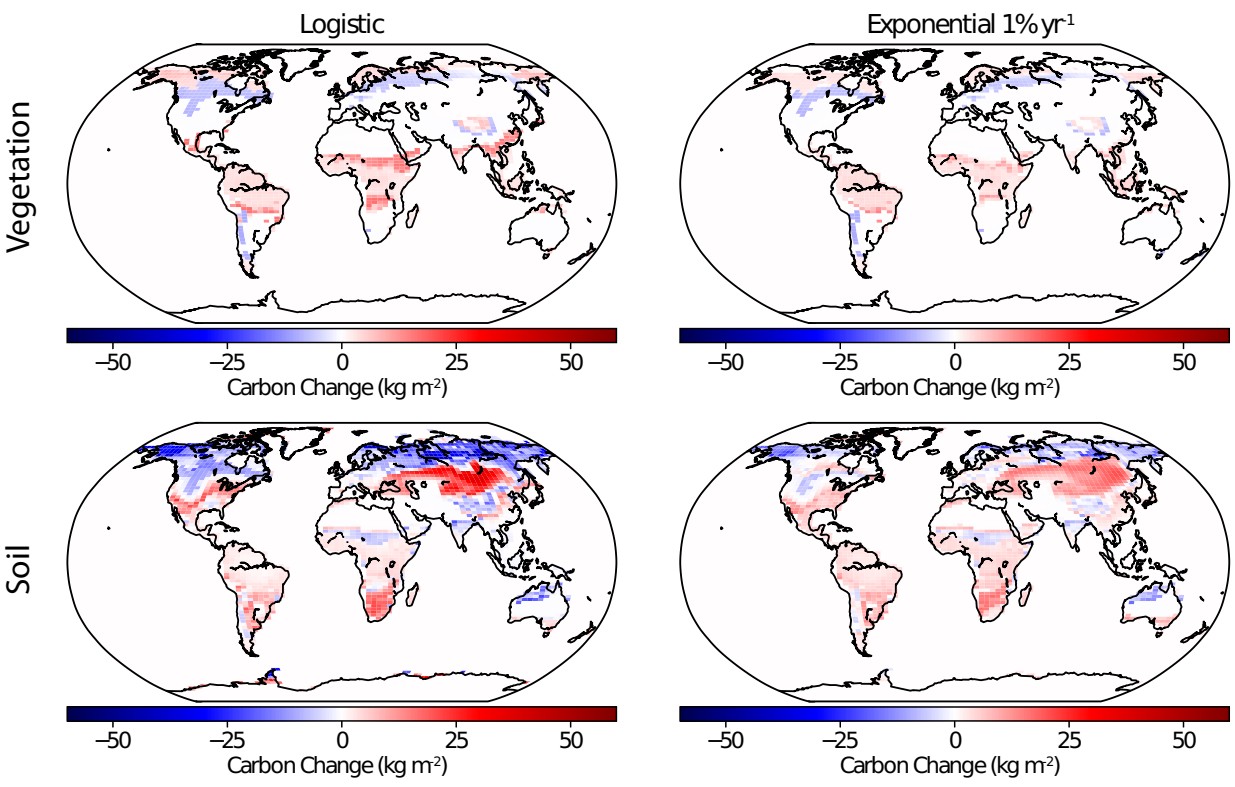

**Figure 5.** Change in carbon content (per unit area) between pre-industrial conditions and $2\times$ CO$_2$ concentration for UVic ESCM simulations forced with the logistic and 1% experiments. Top row shows changes in vegetation carbon, bottom row shows changes in soil carbon.

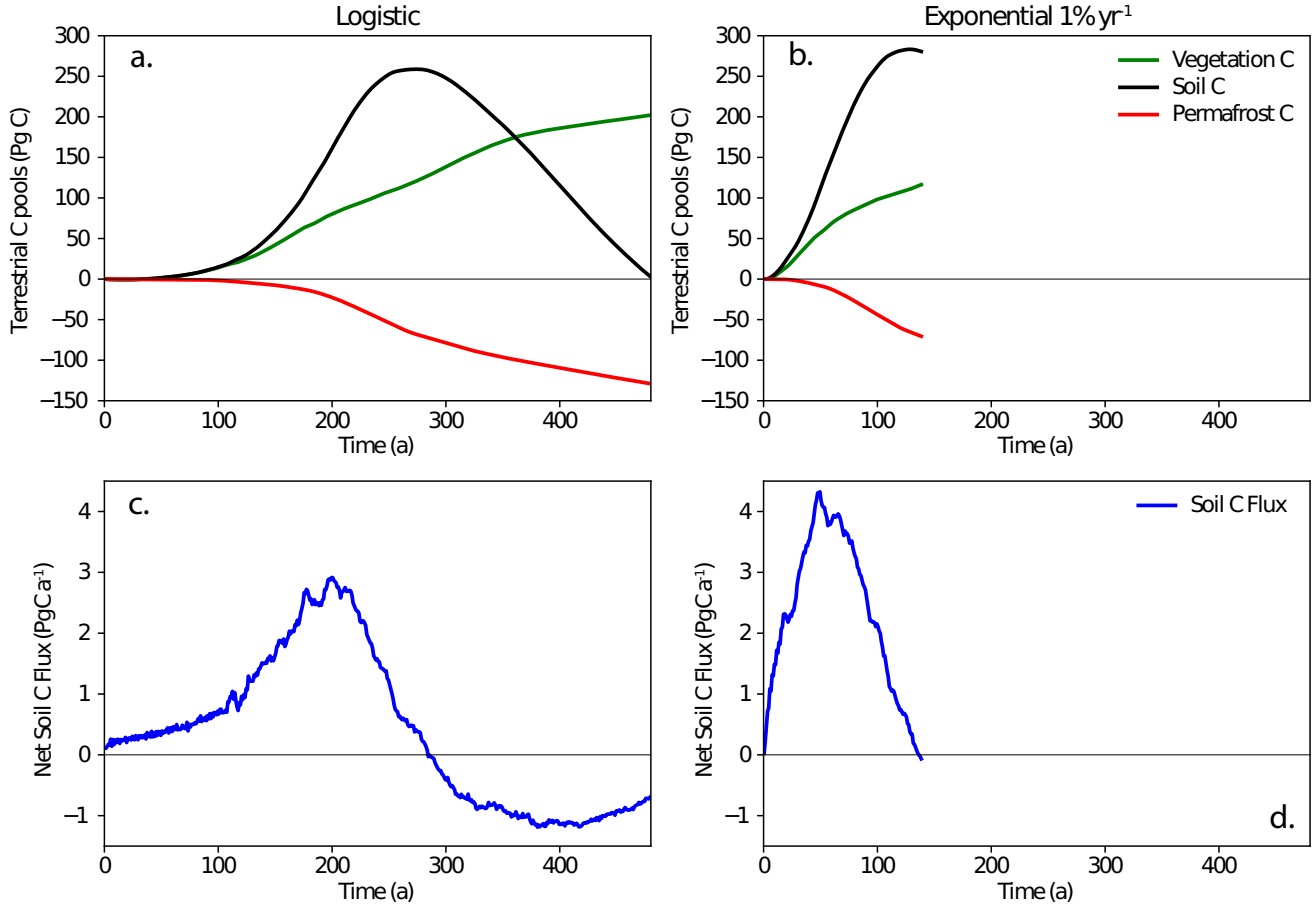

**Figure 6.** a,b – Evolution of terrestrial carbon pool anomalies in the logistic and 1% experiments. c,d – Flux of carbon into the soils computed from litter-fall minus heterotrophic soil respiration (excluding the permafrost carbon pool). Note that in the logistic experiment the enhanced soil respiration feedback overwhelms the increased flux of litter from vegetation leading soils to lose carbon – triggering a carbon sink to source transition on land.

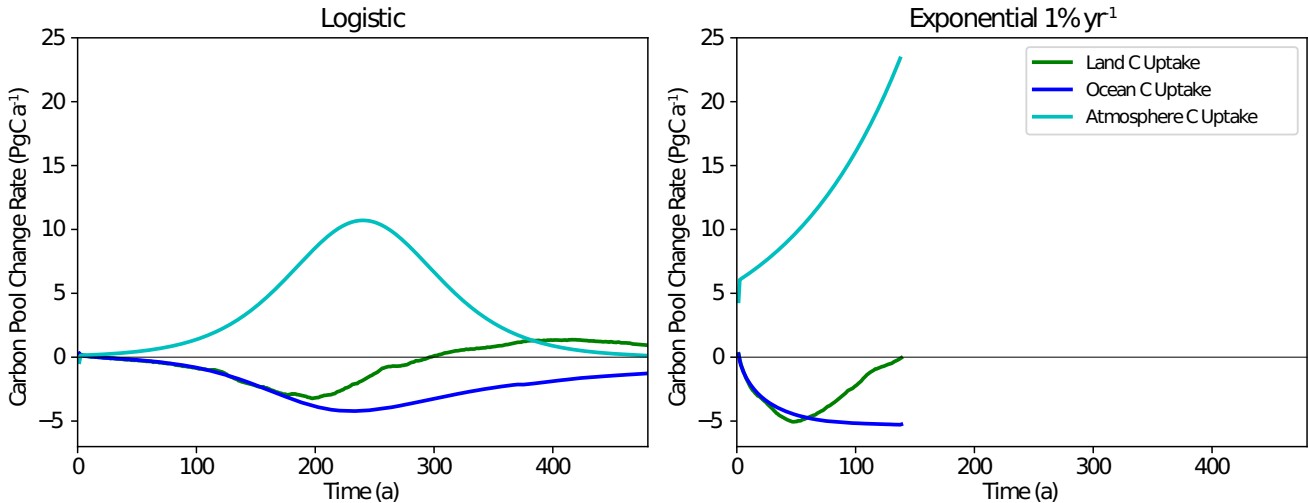

**Figure 7.** Uptake of carbon by the atmosphere, oceans and land systems in the logistic and 1% experiments.

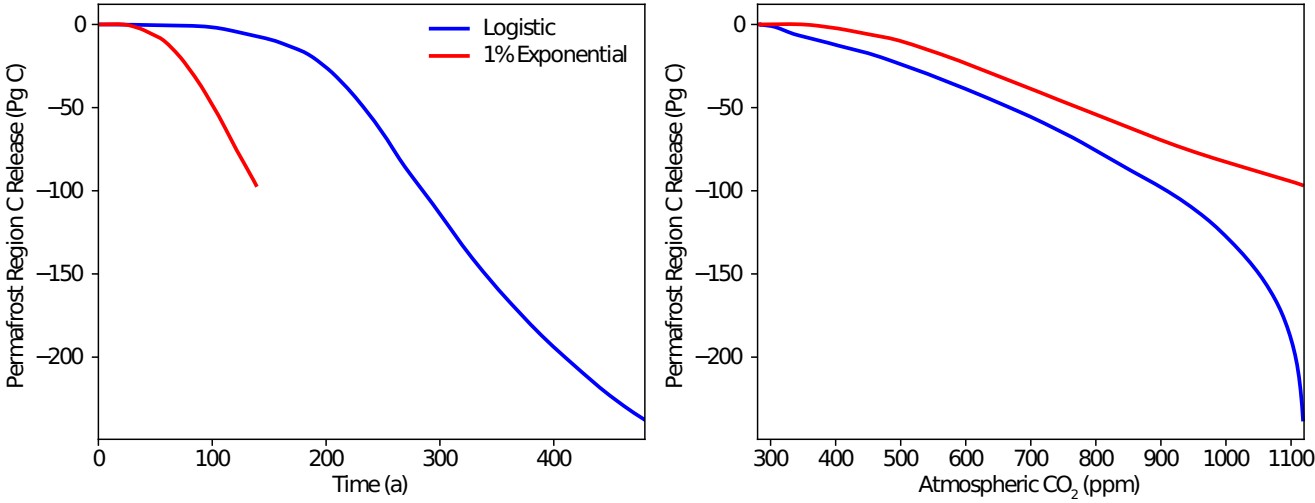

**Figure 8.** Release of carbon from permafrost region under logistic and 1% experiments. Release include both carbon from the permafrost carbon pool and soil carbon from the overlying active layer. Panel **a** compares the simulations with respect to time, and panel **b** with respect to atmospheric $CO_2$ concentration.

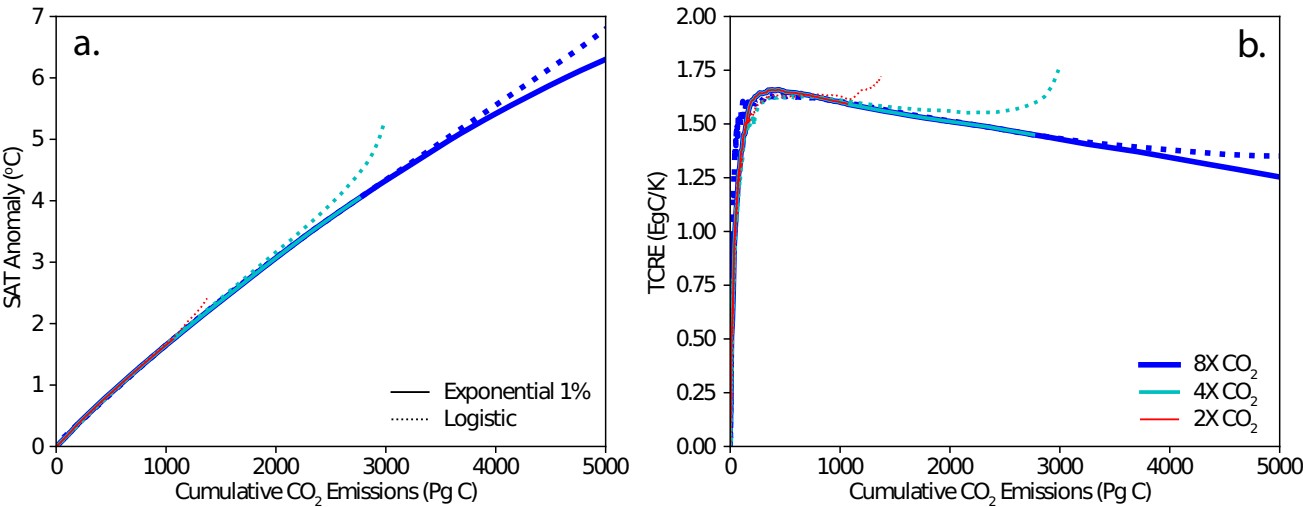

**Figure 9.** (a) Temperature change verses cumulative emissions curve, and (b) TCRE values for all logistic and 1% experiments. Logistic experiments given in dotted lines, 1% experiments given in solid lines. TCRE exhibits strong path-independence except near the end of the logistic experiments when the implied rate of emissions slow.

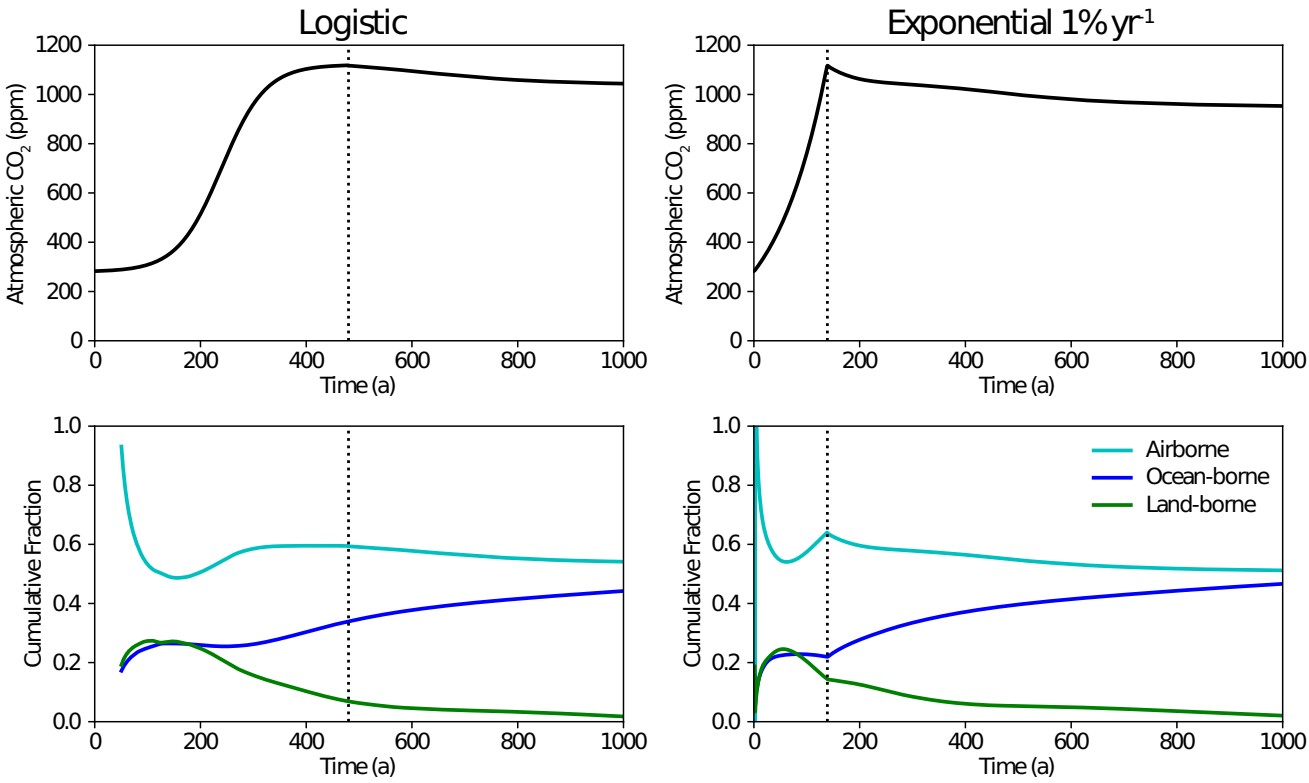

**Figure 10.** Evolution of atmospheric $CO_2$ concentration and cumulative carbon fractions for logistic and 1% experiments followed by net zero $CO_2$ emissions. Vertical dotted lines mark transition from prescribed atmospheric $CO_2$ concentration to zero emissions with free-evolving atmospheric $CO_2$ concentrations. Recall that instantaneous carbon fractions are not defined when emissions are zero (i.e. division by 0).

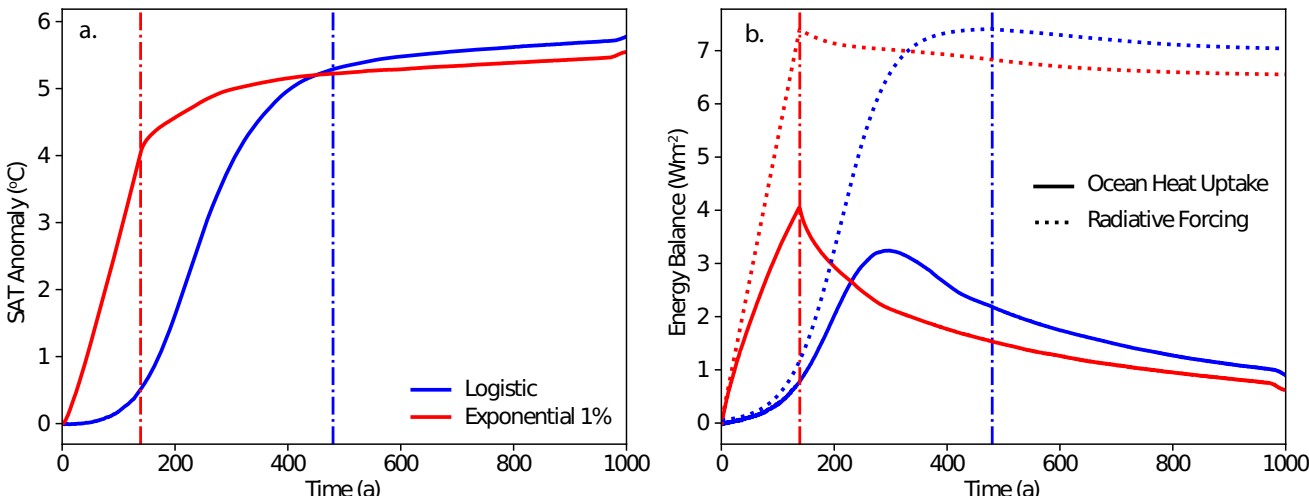

**Figure 11.** (a) Evolution of global Surface Air Temperature (SAT) anomaly during transient run and zero emissions phase of logistic and 1% 4×$CO_2$ experiment. (b) Ocean heat uptake (solid lines) and radiative forcing (dashed lines) for each experiment. Vertical dash-dot lines mark transition from prescribed atmospheric $CO_2$ concentration to zero emissions with free-evolving atmospheric $CO_2$ concentrations. Note ocean heat uptake peaks before emissions cease under the logistic experiment.

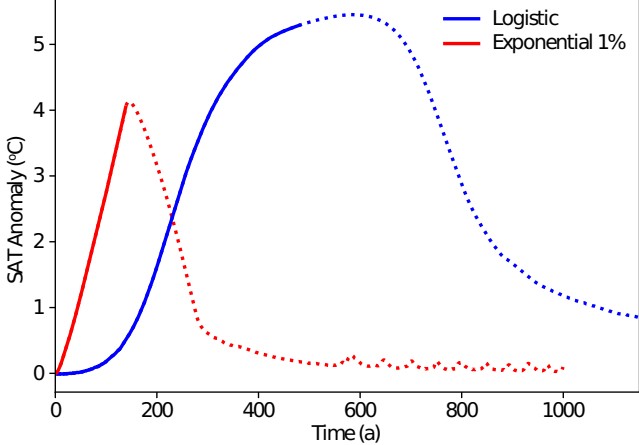

**Figure 12.** Evolution of global Surface Air Temperature (SAT) anomaly during 4×$CO_2$ 1%–up 1%–down and mirrored logistic experiments. Solid lines are the increasing $CO_2$ phase of the simulation and dotted lines are the atmospheric $CO_2$ removal phase of the pathways.

**Table 1.** Parameter values for Logistic $CO_2$ pathways

| Pathway | Rate of Change $k$ (year)$^{-1}$ | Year of mid-point $t_m$ (year) |
|---|---|---|
| $2 \times CO_2$ | 0.030 | 174.7 |
| $4 \times CO_2$ | 0.024 | 239.7 |
| $8 \times CO_2$ | 0.023 | 285.4 |