# Peer review of "Limitations of the 1% experiment as the benchmark idealized experiment for carbon cycle intercomparison in C4MIP"

_Geoscientific Model Development, 2018_

## Referee Comment (RC1) · C.D. Jones (Referee) · 29 Jul 2018

Review of "Limitations of the 1% experiment as the benchmark idealised experiment for carbon cycle inter comparison in C4MIP", by Andrew MacDougall.

This is a well written, clear description of a proposed alternative experiment to the now-standard 1% experiments often used to quantify and compare carbon cycle feedbacks in coupled climate carbon cycle models (so-called C4MIP experiments).

I found this a useful and thoughtful paper which makes some very salient comments about existing experimental design and offers some insights into the limitations of the

standard experiments compared to new "logistic" CO2 pathways. The paper show cases the new pathways using the UVIC EMIC.

In general, both personally and as a co-chair of C4MIP, I find this level of analysis and engagement very pleasing to see, and it will certainly help drive the further evolution of C4MIP in the future (I'm not yet ready to think about CMIP7 though!). C4MIP is explicitly aimed at ESMs, although we welcome EMIC participation. But perhaps for a next generation we should more explicitly engage with EMICs and provide additional simulations which EMICs can lead on to supplement joint ESM/EMIC runs. In fact it was a requirement of CMIP6 that no MIPs added new experiments which had not been tried by at least some models. They (very reasonably) wanted to avoid too many brand new experiments being suggested and possibly wasting time of model groups. So it is really positive to see suggestions like this also being tested with a model.

I list below some comments which I hope will be useful both for the improvement of this manuscript and also in general as part of the evolving discussion. There are some areas of literature which can be helpful, and there are some issues which are relevant to ESMs more than EMICs (mainly around computational expense). But overall I very much like this paper and would recommend publication with only minor amendments.

My main question really is not just the choice of scenario - what do you recommend about an analysis technique. You do not mention performing coupled/uncoupled simulations with the logistic pathway - so how would you look at climate-carbon and CO2-carbon feedbacks? Would you still want to do COU, BGC and RAD versions of the logistic pathway? (which would increase computational cost of course). How do these metrics (beta/gamma) evolve in time?

or are you suggesting keeping the 1% run for the feedback separation and using the logistic run to look more at emissions/TCRE/AF?

It would be good to be clear on the intended USE as well as scenario that you are suggesting.
[Figure]

Otherwise, I list some comments below which I hope you find useful. It would be great to involve you in future discussions around C4MIP analysis and experimental design.

Chris Jones

1. In several places, including the abstract and conclusions the paper mixes up features of the models/results with features of the experiment itself. For example you say sink-to-source transition is "absent from the 1% experiment". I think you should be a bit stricter in which phrasing you use - the sink-to-source transition is neither present nor absent in the 1% experiment - but it will depend on the results. It may or may not occur depending on the model. You might be able to say it is more likely in one set of model runs than another, but it is not a "feature of the experiment".

2. The paper gives a nice overview of the history of the 1% simulation. There has, though, been more discussion around the choice of this for C4MIP than acknowledged here (it's not true to say, "a clear rationale for... 1% experiment... is absent". The best paper on this is Gregory et al (2009, J.Climate). They look in some detail at the Friedlingstein 2006 paper and discuss some of the limitations you mention. They conclude that the 1% should be used and cumulative airborne fraction is a good measure. This is closely related to subsequent papers which derived TCRE or similar metrics relating cumulative emissions to warming levels. Gregory et al also perform and acknowledge differences between scenarios due to rate of change - beta and gamma feedback metrics are seen to vary in 0.5%, 1% and 2% rates of rise.

3. I like how you show various outputs change in time during the various simulations (airborne fraction etc). Can you also derive and show TCRE? You may find that this is actually better behaved in terms of being more constant in time and between scenarios. Which is a nice feature of it in fact.

4. Top of page 5 lists a nice sequence of phases (accelerating/decelerating emissions etc). I agree it is good to make sure these are assessed. In fact RCP2.6 makes a nice example of this succession and my 2016 ERL paper

(http://iopscience.iop.org/article/10.1088/1748-9326/11/9/095012 ). In there we show explicitly a sequence of how human and nature sinks/sources gradually transition from positives to negatives and the interesting dynamics of the earth system. To some extent therefore this scenario can achieve (but not in a clean idealised way) they same sequence that you get via your logistic pathway.

5. I don't disagree with your choice of a pathway - it would indeed be useful. There are also many other possible choices which would be useful. Various ones were discussed during our selection of the latest generation of C4MIP experiments, and include:

- 4xCO2 run, BGC mode, extended beyond 150 years - this gives a large signal to noise and the step change helps avoid conflating various timescales of response

- ZEC - as you suggest a sudden stop in emissions and let the model run free - ideally from a "policy relevant" level of CO2 (such as 2xCO2, rather than 4xCO2)

- CO2 pulse (as per Joos et al 2013, ACP)

- 1% ramp-down

- other (faster/slower than 1%) idealised % runs

there were also desires to run other scenarios as well as the idealised cases (e.g. an emissions-driven RCP2.6). We also tried to align with other MIPs - such as LUMIP.

In conclusion therefore - in order to not end up with way too many model years required from model groups, we selected a small and succinct set. It is highly likely, as you suggest, that this is not perfect and there will be value in other simulations too. For CMIP7 we can certainly open this discussion again and evolve our thinking once more.

On reflection I feel the ZEC run in particular would be very valuable.

And in fact the 1% ramp-down has now entered into CMIP6 via CDR-MIP. CDRMIP is explicitly focussed on negative emissions, as the name suggests. Please can you mention this and the negative-pulse experiment discussed in Keller et al (2018, GMD)

[Figure]

So in summary - the main concern over your suggestions is simply computational expense. Your logistic experiment is many hundreds of years - I can value in this, but it needs to be accessible by ESM groups. If we were to require BGC coupled version too then this doubles.

6. Your point about needing to explore low stabilisation and/or peak-and-decline scenarios is well made, and I fully agree. In fact I'd like to point you to my recent phd thesis available here: https://ore.exeter.ac.uk/repository/handle/10871/27943 - this has (I hope!) some useful background on the feedback framework (section 3.1) including discussion of Gregory et al 2009 - then I make some very similar points to you in section 4.3

minor points:

1. Intro. don't confuse CMIP and IPCC - they have very different remits (even if in reality there is overlap of who takes part). CMIP is the modelling community. They design and run the simulations. IPCC assembles experts to assess the literature - these often draw on, but are not limited to, CMIP simulations. IPCC itself neither does, nor recommends science - it does not choose which scenarios for example CMIP should run.

2. 8xCO2 might be interesting, but (hopefully!) is not policy relevant. I think this would stretch any linearity of the system and not be useful for policy targets. I would expect most ESM groups therefore not to do this one, although EMIC groups, less limited by CPU, may well do.

3. I'm not sure of the value of plotting the compatible fossil fuel emissions for either the 1% or logistic scenarios. To me this is not a relevant quantity. I think experiments should EITHER be "realistic" - i.e. follow a plausible scenario to try to derive useful information about how the real world may unfold, OR be "idealised" - i.e. stripped down or simplified in some way to aide understanding of the system. Both have great value, but shouldn't be mixed. The fossil fuel emissions that would be required to follow a scenario are only really an interesting quantity in the first case. In the second case I don't

think they have either scientific nor policy interest. So I would stick to showing more process-based quantities, such as the land/ocean components, the airborne fraction etc. But not the fossil emissions.

4. in figure 6 - as well as a split into soil/veg carbon. Have you also looked at regional splits? e.g. tropics vs high-latitudes? I could imagine these behave differently and might be interesting to see them separated.

5. on p.7 you say that the increasing ocean-fraction has never been pointed out. While this is true of AR5 (perhaps an omission there), the analysis in Jones et al (2013, J. Climate) does cover this - see the bottom right of our figure 7.

6. a couple of other papers you might want to see: Randerson et al 2015, GCB on the long timescales and how the ocean becomes more important; Schwinger et al 2018, GRL, on ocean carbon reversibility.

———————————————————

---

## Referee Comment (RC2) · A.-I. Partanen (Referee) · 1 Nov 2018

First, I want to disclose that I received a draft of the manuscript before it was published in GMDD and my summer intern used the provided scenarios in our research that is hopefully published at some later time. Although this can be perceived as a minor conflict of interest, I have currently no plans to collaborate with Dr. MacDougall and believe I can deliver an impartial review of the manuscript.

The manuscript describes a new idealized scenario that could be used in C4MIP carbon cycle model intercomparison and potentially replace the standard 1% scenario. The author has also conducted model simulations with the UVic ESCM, an Earth system model of intermediate complexity, to study how the proposed scenarios compare with the 1% scenario. He presents convincing arguments of the limitations of the 1% scenario and how the proposed scenarios could address these. Thus the scenarios presented are potentially an important contribution to model intercomparisons. In addition, they can be valuable in single-model studies as well when idealized scenarios are needed. In our case, we needed a CO2-only scenario that would be of similar length as a historical (1750-2005) + RCP run until 2100. 1% scenario was impractical because the CO2 concentration increase is so much faster compared to the historical scenario. The manuscript is clearly structured and mostly clearly written, although I would prefer more punctuation, as some sentences are hard to read. I recommend the manuscript to be published with some minor improvements.

The other reviewer had many good comments and suggestions, and I agree on almost all of them. One exception is that I think that diagnosed emissions should be shown as they are now. They are used in the discussion of the results and are an important part of understanding the source-sink transitions for example.

Minor comments:

I do not know the conventions, but I would consider using increasing and decreasing emissions instead of accelerating and decelerating emissions. I think that would be more clear. "Accelerating" could be potentially interpreted (or at least misinterpreted) that the rate of change of emissions is increasing, but you seem to imply only that emissions are higher on year n than on year n-1.

Page 1, Line 6: I agree with the first reviewer that it should be made more clear when you are talking about the experiment design and when about the results of the simulations.

Page 1, Line 20: I'm aware that there are several ways to spell out what TCRE stands for. Gillet et al. (2013) used "Transient climate response to cumulative carbon emissions". H. D. Matthews recommended to use "Transient climate response to cumulative

[Figure]

CO2 emissions" (personal communication, 2016). Although the form used here is consistent with the abbreviation (even more than most other versions), I think that including the word carbon or CO2 would be informative here.

Page 4, Line 29: I think these stages are not fully exhaustive. Constant emissions would be at least one easily conceived idealised state of emissions.

Page 5, second lines 3-5 (in 2.3, the line numbering is confusing here): I have been doing some tests with the UVic ESCM by taking restart files from the preindustrial state with prescribed constant CO2 concentration and used them in a zero-emission driven simulation. The sudden transition from concentration driven to emission driven has caused some imbalances in the model's carbon cycle, and the model was not in equilibrium anymore in contrast with my expectations. Therefore, I would guess there might be something similar happening in your case as well when you do the switch to zero emissions. Did you notice anything like that?

Page 5, second lines 6-7: The wording is a bit imprecise here. The negative emission scenario is only the negative emission part of 1% up, 1% down scenario. Maybe there is a way to be precise and keep the sentence still readable?

Page 6, line 16-17: Are the emissions raw model (annual) output or have you applied some running-mean averaging or something similar?

Page 6, line 27: I think the word "near-surface" is somewhat misleading here. It lead me to think whether only surface-ocean is included. I think that deep ocean carbon is farther away from surface than many fossil carbon reservoirs.

Page 6, Line 36 (or 2?): I would use "decrease" instead of slow. This sentence is also an example that would be more readable with a comma (As emissions slow, the land system. . .). Without comma, the beginning of the sentence could be misunderstood so that emissions slow the land system.

Page 6, Line 37 (or 3): What do you mean with "net emissions" here? Diagnosed fossil

fuel emissions or are you taking into account the carbon released from land? If the former, the sentence might be more clear without the word "net". If the latter (which I doubt), it should be written explicitly.

Page 7, Line 12: I would recommend replacing "not captured" with "not visible" or "not present" or something similar. To me, "not captured" sounds like a phrase you would use when a model cannot capture some process due to lack of relevant physical description.

Page 7, line 14: I would refer to Fig. 5c here.

Page 7, line 27: It's probably clear to most readers, but I think it would be better to avoid the potential interpretation that slower warming itself is the cause when the cause is approximately that for a given warming, the longer simulation releases more carbon. Thus, I would rephrase the sentence. Page 7, second Line 8: I think the "when emissions cease" could be interpreted also to mean "after emissions cease". Could you make it clearer that you are referring to the very moment of transition (e.g. at the turning point).

Page 8, Line 15: Can you explain the difference in ZEC between the experiments? Ehlert and Zickfeld (2017) would probably be a good reference here.

Page 10, Lines 21 and second 8: I would recommend using "I" instead of "we" in single-author paper.

Figure 2. This figure is basically replicating part of Figure 4, right? Is it necessary redundancy? Also, the lines are quite hard to read due to overlapping. Could you at least on the 1% side divide the line to 2X,4x,and 8x parts and say that 8x includes also the other two. I know that correcting this and keeping all the figures looking consistent is hard, but especially in Fig. 2b and d it's hard to tell the lines apart.

Figures in general: The style is not entirely consistent. In some figures, the time axis is only to the end of the simulations while in others there is some empty space which I

see no reason for (especially in Fig. 5bdf.)

Technical corrections: Page 4, Line 18: UVic ESCM

Page 5, Line 20 and elsewhere: I'm not sure of journal style, but I think normally you should capitalize "Figure" when coupled with a number.

Page 7, Line 32: Remove either "will" or "s"'s in increases, deceases. . .

Figure 1 caption: add missing % after 0.46

REFERENCES Ehlert, D. & Zickfeld, K. What determines the warming commitment after cessation of CO 2 emissions? Environ. Res. Lett. 12, 015002 (2017).

---

## Author Comment (AC1) · 26 Nov 2018

**Response to reviews of manuscript:**

Limitations of the 1% experiment as the benchmark idealized experiment for carbon cycle intercomparison in C⁴MIP

 I appreciate the thoughtful comments of both reviewers and have responded to each comment below. The reviews are copied verbatim and are italicized. Author responses are in regular font. Changes made to the manuscript are blue.

**Response to Reviewer 1:**

*Review of "Limitations of the 1% experiment as the benchmark idealised experiment for carbon cycle inter comparison in C4MIP", by Andrew MacDougall.*

*This is a well written, clear description of a proposed alternative experiment to the now standard 1% experiments often used to quantify and compare carbon cycle feedbacks in coupled climate carbon cycle models (so-called C4MIP experiments).*

*I found this a useful and thoughtful paper which makes some very salient comments about existing experimental design and offers some insights into the limitations of the standard experiments compared to new "logistic" CO2 pathways. The paper show cases the new pathways using the UVIC EMIC.*

*In general, both personally and as a co-chair of C4MIP, I find this level of analysis and engagement very pleasing to see, and it will certainly help drive the further evolution of C4MIP in the future (I'm not yet ready to think about CMIP7 though!). C4MIP is explicitly aimed at ESMs, although we welcome EMIC participation. But perhaps for a next generation we should more explicitly engage with EMICs and provide additional simulations which EMICs can lead on to supplement joint ESM/EMIC runs. In fact it was a requirement of CMIP6 that no MIPs added new experiments which had not been tried by at least some models. They (very reasonably) wanted to avoid too many brand new experiments being suggested and possibly wasting time of model groups. So it is really positive to see suggestions like this also being tested with a model.*

*I list below some comments which I hope will be useful both for the improvement of this manuscript and also in general as part of the evolving discussion. There are some areas of literature which can be helpful, and there are some issues which are relevant to ESMs more than EMICs (mainly around computational expense). But overall I very much like this paper and would recommend publication with only minor amendments.*

*My main question really is not just the choice of scenario - what do you recommend about an analysis technique. You do not mention performing coupled/uncoupled simulations with the logistic pathway - so how would you look at climate-carbon and CO2- carbon feedbacks? Would you still want to do COU, BGC and RAD versions of the logistic pathway? (which would increase computational cost of course). How do these metrics (beta/gamma) evolve in time?*

*Or are you suggesting keeping the 1% run for the feedback separation and using the logistic run to look more at emissions/TCRE/AF?*

*It would be good to be clear on the intended USE as well as scenario that you are suggesting.*

*Otherwise, I list some comments below which I hope you find useful. It would be great to involve you in future discussions around C4MIP analysis and experimental design.*

*Chris Jones*

I did conduct simulations with Radiative $CO_2$ and Biogeochemically coupled $CO_2$ under the logistic 4X $CO_2$ experiment to allow comparison of Beta and Gamma metrics.

[Figure]

What these simulations ended up showing was the Beta and Gamma metrics are scenario dependent and evolve in time, points that have been clearly articulated in existing literature (e.g. Arora et al. 2013). Since these experiments added so little to the conclusions of the paper I decided to not include the experiments in the submitted manuscript.

To clearly state the intended use of the logistic experiment a new subsection has been added to the discussion section of the paper. The subsection reads:

**"4.2 Recommendations for incorporation of the logistic experiment into CMIP7**

The key advantages of the logistic experiment over the 1% experiment are that the logistic experiment captures the phase of declining emissions, and allows for a smoother transition to zero emissions or negative emissions scenarios. The principle disadvantages of switching to the logistic experiment are the much higher computation cost of the experiment, and the loss of historical continuity in experiment design. Therefore several options are available for incorporation of the logistic experiment into the CMIP7 iteration of C[4]MIP, ranging from full replacement of the 1% experiment, to incorporation of the logistic experiment into the Tier 2 experiment recommendations.

Given the high computation cost of feedback separation, which necessitates running fully coupled, radiatively coupled, and biogeochemically coupled simulations (Gregory et al., 2009; Arora et al., 2013), it is prohibitively expensive for the logistic experiment to be used

for this purpose. Thus I recommend that the logistic experiment be added to the Tier 1 set of experiments, in addition to the 1% experiment. With the logistic experiment used for examining carbon fraction under declining emissions, and evaluating the strength of the permafrost carbon cycle feedback. Either the 2X $CO_2$ or 4X $CO_2$ versions of the logistic experiment could be incorporated into the Tier 1 experiments, with the 2X $CO_2$ version having the advantage of being 'policy relevant', and less computationally demanding."

*1. In several places, including the abstract and conclusions the paper mixes up features of the models/results with features of the experiment itself. For example you say sink to- source transition is "absent from the 1% experiment". I think you should be a bit stricter in which phrasing you use - the sink-to-source transition is neither present nor absent in the 1% experiment - but it will depend on the results. It may or may not occur depending on the model. You might be able to say it is more likely in one set of model runs than another, but it is not a "feature of the experiment".*

The abstract and conclusions have been re-written to clearly distinguish the experiment from the results of the experiment in the UVic ESCM. The abstract did read:

[revised manuscript text omitted]

*2. The paper gives a nice overview of the history of the 1% simulation. There has, though, been more discussion around the choice of this for C4MIP than acknowledged here (it's not true to say, "a clear rationale for… 1% experiment… is absent". The best paper on this is Gregory et al (2009, J.Climate). They look in some detail at the Friedlingstein 2006 paper and discuss some of the limitations you mention. They conclude that the 1% should be used and cumulative airborne fraction is a good measure. This is closely related to subsequent papers which derived TCRE or similar metrics relating cumulative emissions to warming levels. Gregory et al also perform and acknowledge differences between scenarios due to rate of change - beta and gamma feedback metrics are seen to vary in 0.5%, 1% and 2% rates of rise.*

Thanks for the citation. The sentence: "However, a clear rationale for using the 1% experiment for analysis of the carbon cycle is absent from the literature." has been deleted.

A brief summary of the rational given in Gregory et al. 2009 has been incorporated into the history of the 1% experiment.

The lines: "However, later studies utilizing model output from the Coupled Climate--Carbon Cycle Model Intercomparison Project (C$^4$MIP) implicitly criticized the choice of the A2 scenario, calling for the 1% experiment to be used in place of a modified scenario (Matthews et al. 2009). This recommendation was implemented with the CMIP5 protocols calling for benchmark carbon cycle experiments to be carried out using a 1% experiment (Taylor et al 2012)."

Have been changed to:

"However, later studies utilizing model output from the Coupled Climate–Carbon Cycle Model Intercomparison Project (C$^4$MIP) implicitly criticized the choice of the A2 scenario (Gregory et al., 2009; Matthews et al., 2009). Gregory et al. (2009) recommended using the

1% experiment in place of a modified scenarios, due to the simplicity of the 1% experiment, the experiment's well established role in model intercomparison projects, and the magnitude of emissions implied by the 1% experiment being of similar magnitude to socioeconomic scenarios. This recommendation was implemented, with the CMIP5 protocols calling for benchmark carbon cycle experiments to be carried out using a 1% experiment (Taylor et al., 2012)."

*3. I like how you show various outputs change in time during the various simulations (airborne fraction etc). Can you also derive and show TCRE? You may find that this is actually better behaved in terms of being more constant in time and between scenarios. Which is a nice feature of it in fact.*

In my experience TCRE is a mess when examining it in time-space instead of cumulative emission-space. The figure below shows cumulative emissions versus temperature change curves, and TCRE values in cumulative emissions-space and time-space.

[Figure]

The figure shows that TCRE is exhibits strong path-independence, with simulations forced with the exponential and logistic experiments exhibiting similar TCRE values, until the emission rate slows in the logistic experiment near the end of the simulations. The strong path-independence is not obvious when plotting TCRE in time-space.

TCRE has been incorporated into the manuscript with a new figure and a paragraph and the end of section 3.1. The figure is:

[Figure]

Figure 9: (a) Temperature change verses cumulative emissions curve, and (b) TCRE values for all logistic and 1% experiments. Logistic experiments given in dotted lines, 1% experiments given in solid lines. TCRE exhibits strong path-independence except near the end of the logistic experiments when the implied rate of emissions slow.

The paragraph is:

"Cumulative emissions versus temperature change curves and TCRE values for the 2X, 4X, and 8X, 1% and logistic experiment simulations are shown in Figure 9. The TCRE relationship in general shows strong independence from forcing scenario, (e.g. MacDougall et al., 2017), a feature which is evident in Figure 9. Near the end of the logistic experiments when the rate of implied CO2 emissions slows, the TCRE values deviate from scenario independence. Theoretical work on the TCRE relationship suggests that path independence should break-down at very high and very low emission rates (MacDougall, 2017). The results shown in Figure 9 and consistent with this understanding. At the time of CO2 doubling the simulated TCRE value is 1.6 EgC K$^{-1}$ under all experiments except the 2X logistic experiment where that value is 1.7 EgC K$^{-1}$. "

*4. Top of page 5 lists a nice sequence of phases (accelerating/decelerating emissions etc). I agree it is good to make sure these are assessed. In fact RCP2.6 makes a nice example of this succession and my 2016 ERL paper (http://iopscience.iop.org/article/10.1088/1748-9326/11/9/095012 ). In there we show explicitly a sequence of how human and nature sinks/sources gradually transition from positives to negatives and the interesting dynamics of the earth system. To some extent therefore this scenario can achieve (but not in a clean idealised way) they same sequence that you get via your logistic pathway.*

Thank you for the link. That paper illustrates well the different stages of emission pathways. A sentence has been added to line 6 of page 5 to acknowledge the prior work done separating the stages of emissions pathway stages. The sentence reads:

"Previous studies have used the multi-gas RCP 2.6 scenario to examine increasing, decreasing, and negative emission stages (Jones et al, 2016b)."

A minor aside here (not that I should be one to complain since my papers are often riddled of minor spelling errors) but my name is misspelled in the citations of Jones et al. 2016b. 'MacDougall' not 'MacDougal'.

*5. I don't disagree with your choice of a pathway - it would indeed be useful. There are also many other possible choices which would be useful. Various ones were discussed during our selection of the latest generation of C4MIP experiments, and include:*

*- 4xCO2 run, BGC mode, extended beyond 150 years - this gives a large signal to noise and the step change helps avoid conflating various timescales of response*

*- ZEC - as you suggest a sudden stop in emissions and let the model run free – ideally from a "policy relevant" level of CO2 (such as 2xCO2, rather than 4xCO2)*

*- CO2 pulse (as per Joos et al 2013, ACP)*

*- 1% ramp-down*

*- other (faster/slower than 1%) idealised % runs*

*there were also desires to run other scenarios as well as the idealised cases (e.g. an emissions-driven RCP2.6). We also tried to align with other MIPs - such as LUMIP.*

*In conclusion therefore - in order to not end up with way too many model years required from model groups, we selected a small and succinct set. It is highly likely, as you suggest, that this is not perfect and there will be value in other simulations too. For CMIP7 we can certainly open this discussion again and evolve our thinking once more.*

*On reflection I feel the ZEC run in particular would be very valuable. And in fact the 1% ramp-down has now entered into CMIP6 via CDR-MIP. CDRMIP is explicitly focussed on negative emissions, as the name suggests. Please can youmention this and the negative-pulse experiment discussed in Keller et al (2018, GMD)*

A citation to Keller et al. (2018) has been added to the description of the negative emission experiments. In line 8 of Section 2.3 the following sentence has been added:

"The 1%–up 1%–down experiment has been incorporated as a standard model experiment for CMIP6 as part of the Carbon Dioxide Removal (CDR) MIP (Keller et al., 2018)."

*So in summary - the main concern over your suggestions is simply computational expense. Your logistic experiment is many hundreds of years - I can value in this, but it needs to be accessible by ESM groups. If we were to require BGC coupled version to then this doubles.*

The issue on computational expense is now discussed in section 4.2. See above response for the text of the new subsection.

*6. Your point about needing to explore low stabilisation and/or peak-and-decline scenarios is well made, and I fully agree. In fact I'd like to point you to my recent PhD thesis available here: https://ore.exeter.ac.uk/repository/handle/10871/27943 - this has (I hope!) some useful background on the feedback framework (section 3.1) including discussion of Gregory et al 2009 - then I make some very similar points to you in section*

Congratulations of the PhD, and thanks for the link. A citation to the thesis has been added following the sentence: "Thus examining the behaviour of the carbon cycle under conditions of decelerating emissions is a area of imminent importance (e.g. Jones, 2017)."

*minor points:*

*1. Intro. don't confuse CMIP and IPCC - they have very different remits (even if in reality there is overlap of who takes part). CMIP is the modelling community. They design and run the simulations. IPCC assembles experts to assess the literature - these often draw on, but are not limited to, CMIP simulations. IPCC itself neither does, nor recommends science - it does not choose which scenarios for example CMIP should run.*

The sentence:

"In the Fifth Assessment Report of the IPCC"

Has been changed to:

"In the Fifth phase of the Climate Model Intercomparison Project (CMIP5)"

The sentence: "Two of the idealized experiments outlined by the Climate Model Intercomparison Project (CMIP) and the Intergovernmental Panel on Climate Change (IPCC)", has been changed to:

"Two of the idealized  experiments outlined by the Climate Model Intercomparison Project (CMIP)"

The sentence:

"In preparation for the sixth assessment report of the IPCC (AR6) …"

Has been changed to:

"In preparation for CMIP6 …"

*2. 8xCO2 might be interesting, but (hopefully!) is not policy relevant. I think this would stretch any linearity of the system and not be useful for policy targets. I would expect most ESM groups therefore not to do this one, although EMIC groups, less limited by CPU, may well do.*

The atmospheric $CO_2$ concentration at the end of the 8x $CO_2$ experiments is 2240ppm. The UVic ESCM forced with the 8x logistic experiment diagnoses emissions of 5580 PgC. A total less than half of the 12500 PgC estimated to be in the Total Recoverable Base of fossil fuel reserves (e.g. Swart & Weaver, 2012). The RCP 8.5 extension to 2300 has final $CO_2$ concentrations of 1980 ppm (Meinshausen et al 2011). Thus, while I agree that the 8x $CO_2$

experiment should not be the primary idealized experiment used for model intercomparison, I do not share your optimism about the implausibility of the pathway

*3. I'm not sure of the value of plotting the compatible fossil fuel emissions for either the 1% or logistic scenarios. To me this is not a relevant quantity. I think experiments should EITHER be "realistic" - i.e. follow a plausible scenario to try to derive useful information about how the real world may unfold, OR be "idealised" - i.e. stripped down or simplified in some way to aide understanding of the system. Both have great value, but shouldn't be mixed. The fossil fuel emissions that would be required to follow a scenariomare only really an interesting quantity in the first case. In the second case I don't think they have either scientific nor policy interest. So I would stick to showing more process-based quantities, such as the land/ocean components, the airborne fraction etc. But not the fossil emissions.*

Reviewer 2 disagreed with this point and requested that the diagnosed emission be left in the figures. I have thus left the diagnosed emission in the figures.

*4. in figure 6 - as well as a split into soil/veg carbon. Have you also looked at regional splits? e.g. tropics vs high-latitudes? I could imagine these behave differently and might be interesting to see them separated.*

A new figure has been created to complement Figure 6. The figure examines the regional changes in the vegetation and soil carbon pools between pre-industrial conditions and 2X $CO_2$ conditions.

[Figure]

Figure 5. Change in carbon content (per unit area) between pre-industrial conditions and 2X CO2 concentration for UVic ESCM simulations forced with the logistic and 1% experiments. Top row shows changes in vegetation carbon, bottom row shows changes in soil carbon.

A paragraph describing the figure has been added to the manuscript below line 12 of page 7 to describe the figure:

"Figure 5 displays the change in vegetation and soil carbon between pre-industrial conditions and the time of doubled atmospheric $CO_2$ under the 4X $CO_2$ 1% and logistic experiments. The spatial patterns of change are similar under both experiments, but of greater magnitude under the logistic experiment. Vegetation experiences a loss of carbon in the Andes and in mid-latitude northern extra-tropics, while gains in vegetation carbon are seen in the topics, subtropics, sub-arctic and arctic regions. Soils show a reduction in carbon in the permafrost region, boreal forests, and Sahel. Increases in soil carbon in seen in central North America, central Eurasia, and southern Africa, regions generally corresponding to grasslands. Overall the figure shows complex biome-specific responses of the terrestrial biosphere to increasing atmospheric $CO_2$ concentration."

*5. on p.7 you say that the increasing ocean-fraction has never been pointed out. While this is true of AR5 (perhaps an omission there), the analysis in Jones et al (2013, J. Climate) does cover this - see the bottom right of our figure 7.*

The phrase: "but to our knowledge increasing ocean-borne fraction under decelerating emissions has not been explicitly pointed out in literature."

has been deleted and has been replaced by:

",and is evident for model simulations under the peak-and-decline RCP 2.6 scenario in CMIP5 ESM output (Jones et al., 2013). "

*6. a couple of other papers you might want to see: Randerson et al 2015, GCB on the long timescales and how the ocean becomes more important; Schwinger et al 2018, GRL, on ocean carbon reversibility.*

Citations to both papers have been incorporated into the manuscript.

**Response to Reviewer 2:**

*First, I want to disclose that I received a draft of the manuscript before it was published in GMDD and my summer intern used the provided scenarios in our research that is hopefully published at some later time. Although this can be perceived as a minor conflict of interest, I have currently no plans to collaborate with Dr. MacDougall and believe I can deliver an impartial review of the manuscript.*

*The manuscript describes a new idealized scenario that could be used in C4MIP carbon cycle model intercomparison and potentially replace the standard 1% scenario. The author has also conducted model simulations with the UVic ESCM, an Earth system model of intermediate complexity, to study how the proposed scenarios compare with the 1% scenario. He presents convincing arguments of the limitations of the 1% scenario and how the proposed scenarios could address these. Thus the scenarios presented are potentially an important contribution to model intercomparisons. In addition, they can be valuable in single-model studies as well when idealized scenarios are needed. In our case, we needed a CO2-only scenario that would be of similar length as a historical (1750-2005) + RCP run until 2100. 1% scenario was impractical because the CO2 concentration increase is so much faster compared to the historical scenario. The manuscript is clearly structured and mostly clearly written, although I would prefer more punctuation, as some sentences are hard to read.*

I have edited the manuscript to try to improve punctuation and readability.

*I recommend the manuscript to be published with some minor improvements.*

*The other reviewer had many good comments and suggestions, and I agree on almost all of them. One exception is that I think that diagnosed emissions should be shown as they are now. They are used in the discussion of the results and are an important part of understanding the source-sink transitions for example.*

The diagnosed emission have been retained in the revised version of the manuscript.

*Minor comments:*

*I do not know the conventions, but I would consider using increasing and decreasing emissions instead of accelerating and decelerating emissions. I think that would be more clear. "Accelerating" could be potentially interpreted (or at least misinterpreted) that the rate of change of emissions is increasing, but you seem to imply only that emissions are higher on year n than on year n-1.*

Throughout the manuscript 'accelerating' has been changed to 'increasing', and 'decelerating' to 'declining'.

*Page 1, Line 6: I agree with the first reviewer that it should be made more clear when you are talking about the experiment design and when about the results of the simulations.*

See response to Reviewer 1. The abstract and conclusions have been re-written to make clear the distinction between the experiment design, and the results of the simulations with the UVic ESCM.

*Page 1, Line 20: I'm aware that there are several ways to spell out what TCRE stands for. Gillet et al. (2013) used "Transient climate response to cumulative carbon emissions". H. D. Matthews recommended to use "Transient climate response to cumulative CO2 emissions" (personal communication, 2016). Although the form used here is consistent with the abbreviation (even more than most other versions), I think that including the word carbon or CO2 would be informative here.*

Leaving out 'CO$_2$' was an unintentional error. I have in the past used the "Transient Climate Response to CO$_2$ Emissions" definition (e.g. MacDougall, 2015). This error has been corrected in the revised manuscript.

*Page 4, Line 29: I think these stages are not fully exhaustive. Constant emissions would be at least one easily conceived idealised state of emissions.*

Yes, some studies have used idealized constant rate of emission experiments (e.g. Krasting et al. 2014). This is now acknowledged in the manuscript. A sentences has been added to Page 5 line 6 reading:

"Some studies have used a fifth stage of the emissions pathway where the emission rate is constant (e.g. Krasting et al., 2014). Although useful for some examining some problems, such a state not necessary to capture the likely evolution of CO$_2$ emissions."

*Page 5, second lines 3-5 (in 2.3, the line numbering is confusing here): I have been doing some tests with the UVic ESCM by taking restart files from the preindustrial state with prescribed constant CO2 concentration and used them in a zero-emission driven simulation. The sudden transition from concentration driven to emission driven has caused some imbalances in the model's carbon cycle, and the model was not in equilibrium anymore in contrast with my expectations. Therefore, I would guess there might be something similar happening in your case as well when you do the switch to zero emissions. Did you notice anything like that?*

I re-ran the 2X CO$_2$ 1% transition to zero emissions experiment with Global Sums turned on to check mass conservation. Just before the transition to zero emissions totals are (3[rd] column is carbon):

Total heat (in Joules referenced to 0 C and no ice or snow) and fresh water (in kg)
Total ocean fresh water is the equivalent difference from the ocean volume referenced to socn
 t atm     0.11719870007928E+24 J   0.13077251701155E+17 kg   0.11853186632084E+16 kg
 t snow   -0.16578942267172E+23 J   0.49637551698120E+17 kg   0.00000000000000E+00 kg
 t ice    -0.17937075287971E+22 J   0.53703818227460E+16 kg   0.00000000000000E+00 kg
 t lnd     0.26393241131399E+26 J   0.28829451542494E+18 kg   0.32967334597316E+16 kg
 t ocn     0.24463092309902E+26 J   0.10685384656284E+20 kg   0.37424132403260E+17 kg
 t total   0.50955159491584E+26 J   0.11041764356931E+20 kg   0.41906184526200E+17 kg

Immediately after the transition to zero emissions the sums are (after the model has been restarted with different flags in the mk.in file):

Total heat (in Joules referenced to 0 C and no ice or snow) and fresh water (in kg)

Total ocean fresh water is the equivalent difference from the ocean volume referenced to socn

```
t atm     0.11719870007928E+24 J   0.13077251701155E+17 kg   0.11853186632084E+16 kg
t snow   -0.16578942267172E+23 J   0.49637551698120E+17 kg   0.00000000000000E+00 kg
t ice    -0.17937075287971E+22 J   0.53703818227460E+16 kg   0.00000000000000E+00 kg
t lnd     0.26393241131399E+26 J   0.28829451542494E+18 kg   0.32967334597316E+16 kg
t ocn     0.24463092309902E+26 J   0.10685384656284E+20 kg   0.37424132403260E+17 kg
t total   0.50955159491584E+26 J   0.11041764356931E+20 kg   0.41906184526200E+17 kg
```

Note that the carbon totals are identical on either side of the transition, indicating no error in mass conservation. The model conservers mass to machine precision, as it was designed to do.

I have encounter similar problems with model drift following releases from model spin-up. These are usually either caused by a spin-up that is insufficiently long (10,000 years for the ocean to come to equilibrium is recommended), or slight differences in forcing between the spin-up configuration and the run configuration.

*Page 5, second lines 6-7: The wording is a bit imprecise here. The negative emission scenario is only the negative emission part of 1% up, 1% down scenario. Maybe there is a way to be precise and keep the sentence still readable?*

The referenced sentence did read:

"For the 1% experiment the negative emissions scenario is the 1% up, 1% down experiment used by several previous studies"

And has been changed to:

"The mirrored return negative-emissions-scenario derived from the 1% experiment is the 1%–up 1%–down experiment used by several previous studies"

*Page 6, line 16-17: Are the emissions raw model (annual) output or have you applied some running-mean averaging or something similar?*

These are the raw output, no moving average or filtering has been applied. In my experience with the UVic ESCM inter-annual variability in diagnosed emissions comes from the model overcompensating for non-$CO_2$ forcing such as volcanic eruptions.

*Page 6, line 27: I think the word "near-surface" is somewhat misleading here. It lead me to think whether only surface-ocean is included. I think that deep ocean carbon is farther away from surface than many fossil carbon reservoirs.*

"Near surface" was meant to exclude fossil fuel reserves and ocean sediments. The phrase:

"stored in each of the main near-surface carbon reservoirs"

Has been changed to:

"stored in each of the main fast-cycling carbon reservoirs"

*Page 6, Line 36 (or 2?): I would use "decrease" instead of slow. This sentence is also an example that would be more readable with a comma (As emissions slow, the land system: : :). Without comma, the beginning of the sentence could be misunderstood so that emissions slow the land system.*

The sentence has been re-written from:

"As emissions slow the land system transitions from a carbon sink to a carbon source, while the ocean sink comes to dominate the system absorbing more carbon than the net emissions to the atmosphere."

To:

"As emissions decline, the land system transitions from a carbon sink to a carbon source. The ocean sink comes to dominate the system absorbing more carbon than the anthropogenic $CO_2$ emissions to the atmosphere."

*Page 6, Line 37 (or 3): What do you mean with "net emissions" here? Diagnosed fossil fuel emissions or are you taking into account the carbon released from land? If the former, the sentence might be more clear without the word "net". If the latter (which I doubt), it should be written explicitly.*

As shown above this has been changed to "anthropogenic $CO_2$ emissions" for clarity.

*Page 7, Line 12: I would recommend replacing "not captured" with "not visible" or "not present" or something similar. To me, "not captured" sounds like a phrase you would use when a model cannot capture some process due to lack of relevant physical description.*

"not captured well" has been changed to "not present"

*Page 7, line 14: I would refer to Fig. 5c here.*

A citation to Figure 5c has been added to the end of the sentence.

*Page 7, line 27: It's probably clear to most readers, but I think it would be better to avoid the potential interpretation that slower warming itself is the cause when the cause is approximately that for a given warming, the longer simulation releases more carbon. Thus, I would rephrase the sentence.*

The sentence has been re-written from:

"Figure 8 demonstrates the importance of time in destabilizing permafrost carbon, with the slower warming logistic experiment having a higher release of carbon from permafrost regions at any given $CO_2$ concentration."

To:

"Figure 8 demonstrates the importance of elapsed time in destabilizing permafrost carbon. The logistic experiment implies lower $CO_2$ emission rate and hence a lower rate of warming, results in a higher release of carbon from permafrost regions at any given $CO_2$ concentration. The result is consistent with previous work on the permafrost carbon feedback, which demonstrates a long lag time between forcing and response due to the time taken to thaw soil and decay soil carbon (e.g. Schuur et al., 2015)."

*Page 7, second Line 8: I think the "when emissions cease" could be interpreted also to mean "after emissions cease". Could you make it clearer that you are referring to the very moment of transition (e.g. at the turning point).*

"when emissions cease" has been replaced by: "at the time emission stop"

*Page 8, Line 15: Can you explain the difference in ZEC between the experiments? Ehlert and Zickfeld (2017) would probably be a good reference here.*

An addition panel has been added to old Figure 10 (now Figure X), showing the radiative forcing and ocean heat uptake under both experiments, to explain the difference in ZEC:

[Figure]

Figure 11. (a) Evolution of global Surface Air Temperature (SAT) anomaly during transient run and zero emissions phase of logistic and 1% 4X $CO_2$ experiment. (b) Ocean heat uptake (solid lines) and radiative forcing (dashed lines) for each experiment. Vertical dash-dot lines mark transition from prescribed atmospheric $CO_2$ concentration to zero emissions with free-evolving atmospheric $CO_2$ concentrations. Note ocean heat uptake peaks before emissions cease under the logistic experiment.

A short paragraph explaining the difference in ZEC has been included after line 15 of page 8:

"Figure 11b shows the radiative forcing and ocean heat uptake under both the 1% and logistic ZEC experiments. The figure shows that under the 1% experiment radiative forcing and ocean heat uptake peak the moment emissions cease. While under the logistic experiment ocean heat uptake peaks over a century before emission cease. The declining ocean heat uptake under that logistic experiment explains the smaller ZEC under that

experiment. When emissions cease under the logistic experiment the Earth system is closer thermal equilibrium resulting in a smaller radiative imbalance and unrealized warming. These results are consistent with previous experiments examining the pathway dependence of ZEC (Ehlert and Zickfeld, 2017)."

*Page 10, Lines 21 and second 8: I would recommend using "I" instead of "we" in single-author paper.*

I have changed "we" to "I" in the context of recommendations. "we" is retained when the reader is included in the we.

*Figure 2. This figure is basically replicating part of Figure 4, right? Is it necessary redundancy? Also, the lines are quite hard to read due to overlapping. Could you at least on the 1% side divide the line to 2X,4x,and 8x parts and say that 8x includes also the other two. I know that correcting this and keeping all the figures looking consistent is hard, but especially in Fig. 2b and d it's hard to tell the lines apart.*

Figure 4 does replicate Figure 2. This was done to avoid placing too much emphasis on the negative emission scenarios, which make up only a small part of the paper. However, the paper is now very figure heavy thus Figure 2 has been removed and replaced by Figure 4.

Figure 4 has been re-drafted with a new colour scheme, which makes the lines much easier to distinguish:

[Figure]

*Figures in general: The style is not entirely consistent. In some figures, the time axis is only to the end of the simulations while in others there is some empty space which I see no reason for (especially in Fig. 5bdf.)*

The figures have been re-drafted such that the length of the X-axis is defined by the longest simulation.

*Technical corrections:*

*Page 4, Line 18: UVic ESCM*

Fixed

*Page 5, Line 20 and elsewhere: I'm not sure of journal style, but I think normally you should capitalize "Figure" when coupled with a number.*

Figure is now capitalized when coupled with a number.

*Page 7, Line 32: Remove either "will" or "s"'s in increases, deceases: : :*

Fixed

*Figure 1 caption: add missing % after 0.46*

Fixed.

---

## Author Response (AR2)

*Review by editor:*

*This is really just a technical correction, but I've got into the habit of clicking "minor revision" as only then I can check that the correction has been made before final acceptance. (It is so disappointing if a paper get published with a known glitch!).*

*"3.1 Accelerating & Decelerating Emissions"*

*I assume that this title needs to be changed. I fully agree with reviewer 2 that you need to be completely clear about whether the emissions are actually accelerating/decelerating or only increasing/decreasing.*

Thank you for finding this error. One of the major drawbacks of writing a solo-author paper is not having a second set of eyes to find minor mistakes. I have now gone through the manuscript with the Find function to make sure I have changed all of the accelerating/decelerating to increasing/ decreasing.

*Otherwise this seems like a nice paper. At GMD every model, code or experiment protocol being proposed needs to have been tested (shown to work). It is interesting to me that you have chosen to test your experiment with an EMIC as this might spur others to see whether they get similar results in GCMs sooner rather than later! In this context, it would be appropriate to make your model output publicly available. Would that be possible?*

I have placed the model output into a publically availably directory. The link is: http://wiracocha.stfx.ca:5000/sharing/pZYta8JrV

I have included a sentence in the Data Availability section of the template to refer readers to this link.

"Model output from the 1% and logistic experiments described here are available for download from http://wiracocha.stfx.ca:5000/sharing/pZYta8JrV

"